# Seipin and the membrane-shaping protein Pex30 cooperate in organelle budding from the endoplasmic reticulum

Sihui Wang[1], Fatima-Zahra Idrissi[2,3], Martin Hermansson[4], Alexandra Grippa[2,3], Christer S. Ejsing 🅸 [4,5] & Pedro Carvalho 🅸 [1,2,3]

Lipid droplets (LDs) and peroxisomes are ubiquitous organelles with central roles in eukaryotic cells. Although the mechanisms involved in biogenesis of these organelles remain elusive, both seem to require the endoplasmic reticulum (ER). Here we show that in yeast the ER budding of these structurally unrelated organelles has remarkably similar requirements and involves cooperation between Pex30 and the seipin complex. In the absence of these components, budding of both LDs and peroxisomes is inhibited, leading to the ER accumulation of their respective constituent molecules, such as triacylglycerols and peroxisomal membrane proteins, whereas COPII vesicle formation remains unaffected. This phenotype can be reversed by remodeling ER phospholipid composition highlighting a key function of these lipids in organelle biogenesis. We propose that seipin and Pex30 act in concert to organize membrane domains permissive for organelle budding, and that may have a lipid composition distinct from the bulk ER.

---

[1] Sir William Dunn School of Pathology, University of Oxford, South Parks Road, Oxford OX1 3RE, UK. [2] Cell and Developmental Biology Programme, Centre for Genomic Regulation (CRG), Dr. Aiguader, 88, 08003 Barcelona, Spain. [3] Universitat Pompeu Fabra (UPF), Dr. Aiguader, 88, 08003 Barcelona, Spain. [4] Department of Biochemistry and Molecular Biology, Villum Center for Bioanalytical Sciences, University of Southern Denmark, Campusvej 55, 5230 Odense, Denmark. [5] Cell Biology and Biophysics Unit, European Molecular Biology Laboratory, Heidelberg, Germany. These authors contributed equally: Sihui Wang, Fatima-Zahra Idrissi. Correspondence and requests for materials should be addressed to P.C. (email: pedro.carvalho@path.ox.ac.uk)

The endoplasmic reticulum (ER) produces large amounts of phospholipids and membrane proteins to supply most cellular organelles. Upon their synthesis, many of these molecules are packaged into COPII-coated vesicles to traffic to their final destination[1]. Although the mechanisms of COPII vesicle formation are well characterized, how specific proteins and lipids in the ER give rise to other organelles, such as lipid droplets (LDs) and peroxisomes, remains largely unknown[2]. These are ubiquitous organelles and mutations impairing their functions result in devastating diseases[3,4]. Despite their importance in cell physiology, LD and peroxisome biogenesis at the surface of the ER remain mysterious[2].

In yeast, de novo peroxisome biogenesis involves the budding of membrane-bound pre-peroxisomal vesicles (PPVs) from specific ER domains distinct from COPII assembly sites[5,6]. As cargo, these vesicles were shown to contain certain peroxisomal membrane proteins[5,7–9], such as Pex3, Pex15, and the importomer complex component Pex14, required for the translocation of soluble proteins from the cytoplasm into the peroxisomal matrix[10]. PPV-budding sites are enriched in the membrane-shaping protein Pex30 but their exact composition and mechanism of formation are unknown[5]. In fact, it has been suggested that formation of functional peroxisomes results from the fusion of two distinct types of PPVs formed at the ER[11,12]. However, this latest model is controversial and how PPVs mature to become functional peroxisomes remains ill-defined[2,3,13].

LDs are structurally distinct from peroxisomes consisting of a neutral lipid core, mainly of triacylglycerols (TAGs) and steryl esters, encased in a unique phospholipid monolayer instead of the typical bilayer present in most organelles[14]. The biogenesis of LDs also occurs at specific ER domains but whether and how these relate to PPV-budding sites have not been explored. According to the prevailing model, neutral lipids between bilayer leaflets at specific ER sites coalesce into a lens. The growth of the lens by addition of more neutral lipids results in LD budding into the cytoplasm[14–16]. The growth of a LD at the ER surface is also essential for targeting hairpin-containing membrane proteins to the LD monolayer[17]. Curiously, the LD targeting of at least some hairpin-containing proteins requires factors involved in peroxisomal membrane protein targeting[18], suggesting a link between LD and peroxisome biogenesis.

LD budding sites are marked by seipin, a widely conserved protein frequently mutated in familial forms of lipodystrophy[19–21]. In agreement with its central role at the ER-LD interface, mutations in seipin lead to aberrant LD biogenesis, morphology, and dynamics in a variety of cell types[4,20–25]. Abnormal ER-LD contacts in seipin mutants also reduce efficient targeting of LD-specific proteins and increase the packing defects in the LD monolayer, as detected by the unspecific recruitment of amphipathic helix-containing proteins[22,23,25]. However, factors that cooperate with seipin have not been identified and, as in the case of PPVs, the precise composition of LD budding sites is unclear.

Here we identify Pex30 as a factor cooperating with seipin in the biogenesis of both LDs and PPVs from the ER. We show that Pex30 and seipin are enriched at ER domains where both LDs and PPVs appear to form. Consistent with a functional role at those ER domains, simultaneous deletion of Pex30 and seipin results in strong impairment of LD and peroxisome biogenesis. Strikingly, the organelle budding defect in these mutants can be largely restored by remodeling ER phospholipid composition. These data suggest that seipin and Pex30 organize membrane domains with a lipid composition distinct from the bulk ER, and that is permissive for organelle budding.

**Table 1 Pex30 is enriched in LDs isolated from _fld1Δ_ and _ldb16Δ_ cells**

|  | _fld1Δ_ | | _ldb16Δ_ | |
|---|---|---|---|---|
|  | logFC | _p_-Value | logFC | _p_-Value |
| **Pex30** | **5.06** | $1.4 \times 10^{-5}$ | **4.17** | $4.1 \times 10^{-7}$ |
| Sec61 | 0.02 | $9.6 \times 10^{-1}$ | 1.17 | $3.5 \times 10^{-1}$ |
| Sec63 | 0.09 | $8.4 \times 10^{-1}$ | − 0.12 | $8.7 \times 10^{-1}$ |
| Kar2 | 0.56 | $2.3 \times 10^{-1}$ | 0.62 | $4.0 \times 10^{-1}$ |
| Usa1 | 1.17 | $1.4 \times 10^{-1}$ | 0.47 | $6.3 \times 10^{-1}$ |
| Rtn1 | 1.63 | $1.4 \times 10^{-2}$ | 1.45 | $1.2 \times 10^{-1}$ |
| Rtn2 | 1.60 | $9.6 \times 10^{-3}$ | 1.73 | $2.5 \times 10^{-2}$ |
| Yop1 | 2.35 | $8.2 \times 10^{-4}$ | 1.87 | $2.0 \times 10^{-3}$ |

LD-enriched fractions were analyzed by label-free quantitative proteomics. Besides Pex30 (in bold), abundant ER membrane proteins are shown as comparison

## Results

**Pex30 and seipin cooperate during LD biogenesis.** To identify factors that cooperate with seipin, we hypothesized that they would compensate for its loss and be enriched at LD budding sites in seipin mutants. To test this possibility, fractions enriched in LDs containing also remnants of adjacent but not bulk ER membranes, were isolated from wild-type (WT) and seipin mutant cells, and analyzed by quantitive proteomics[22]. In yeast two polypeptides, Fld1 and Ldb16, form a functional seipin complex[25]. We found that Pex30 was strongly enriched in fractions isolated from _fld1Δ_ and _ldb16Δ_ cells (Table 1 and Supplementary Tables S1, S2). Other abundant ER proteins were not as enriched in these fractions, suggesting that the effect on Pex30 was specific[22]. To validate the proteomics result, we investigated the localization of functional, endogenously expressed Pex30 fused to the fluorescent protein monomeric Neon Green (Pex30-mNG). In WT cells, Pex30-mNG localized non-uniformly throughout the ER as detected by Sec63-mCherry (Fig. 1a), as previously reported[5,6]. In agreement with the proteomics data, _fld1Δ_ and _ldb16Δ_ cells showed dramatic enrichment of Pex30-mNG in LD proximal regions (Fig. 1a and Supplementary Figure 1). This enrichment was observed during all growth stages and was not a consequence of increased Pex30 protein levels, as these were comparable between seipin mutant and WT cells (Fig. 1b). Thus, in the absence of seipin, Pex30 specifically accumulates at ER-LD contacts.

To dissect the links between seipin and Pex30 we combined deletions on both genes. Remarkably, both _pex30Δfld1Δ_ and _pex30Δldb16Δ_ grew much slower than WT cells and the individual deletion mutants. The growth defect was observed at 30 °C and much exacerbated at lower (25 °C) and higher (37 °C) temperatures, both in rich (Fig. 1c) and defined media (Supplementary Figure 2a). Pex30 is the founding member of a larger family of membrane-shaping proteins also including Pex28, Pex29, Pex31, and Pex32[5,6,26,27]. However, mutants with combined deletion of seipin and any other members of the Pex30 family grew mostly at normal rates (Supplementary Figure 2b). Thus, within this protein family, Pex30 appears to have some unique function. Moreover, cells with seipin deletion in conjunction with genes essential for peroxisome biogenesis, such as Pex3 and Pex19, also grew at normal rates, indicating that the defect in _pex30Δfld1Δ_ or _pex30Δldb16Δ_ was not due to a general loss of peroxisomes (Supplementary Figure 2c). Together, these data indicate that enrichment of Pex30 at ER-LD contacts in seipin mutants is functionally relevant.

Pex30 was recently shown to be involved in the budding of PPVs from the ER[5,6]. Given its enrichment at ER-LD contacts in seipin mutants, we wondered whether it could also contribute to

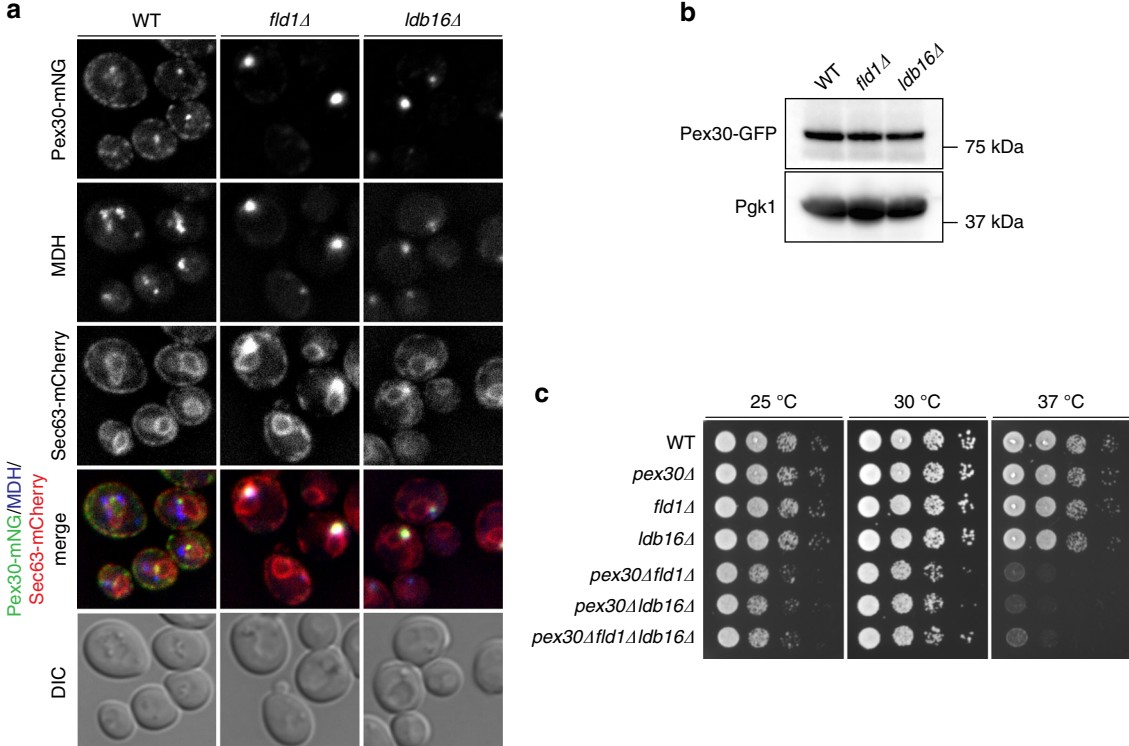

**Fig. 1** Pex30 is localized to LD proximal regions in seipin mutants. **a** Localization of endogenous Pex30 tagged with monomeric Neon Green (Pex30-mNG) in WT, *fld1Δ*, and *ldb16Δ* cells. Endogenous Sec63 was tagged with mCherry and used as an ER marker. LDs were stained with the neutral lipid dye MDH. Bar 2 μm. **b** Levels of endogenous GFP-tagged Pex30 in WT, *fld1Δ*, and *ldb16Δ* cells. Whole-cell extracts were separated by SDS-PAGE and analyzed by western blotting. Pex30-GFP and Pgk1, used as loading control, were detected with anti-GFP and -Pgk1 antibodies, respectively. **c** Tenfold serial dilutions of cells with the indicated genotype were spotted on YPD media and incubated at 25 °C, 30 °C, or 37 °C for 2 days

LD budding. As previously described[20–22,25], *fld1Δ* and *ldb16Δ* mutants displayed abnormally supersized or aggregated LDs, as detected by fluorescence microscopy of cells stained with the neutral lipid-specific dye Bodipy (Fig. 2a). In contrast, LDs in *pex30Δ* mutants appeared largely indistinguishable from WT cells. However, it should be noted that 4% of *pex30Δ* cells (*n* = 300) showed additional Bodipy-positive structures (Supplementary Figure 3a). These were never detected in WT, *fld1Δ*, and *ldb16Δ* cells, and were invariably less bright than LDs and reminiscent of small membranous regions, a hypothesis supported by electron microscopy (EM) (Supplementary Figure 3b). To further explore a potential cooperation between seipin and Pex30, we analyzed the double mutants *pex30Δfld1Δ* and *pex30Δldb16Δ*. Both mutants showed a unique dispersed Bodipy pattern of variable size but often occupying a large fraction of cellular area (Fig. 2a, b). Such aberrant staining was detected in the majority of double mutant cells (Fig. 2b). Moreover, the dispersed Bodipy pattern was frequently scattered with brighter spots of variable sizes. Electron microscopy characterization of the double mutants suggested that the dispersed Bodipy-positive structures correspond to highly convoluted ER membranes (Fig. 2c). Consistent with their staining by Bodipy, these membranous structures are likely enriched in neutral lipids, which do not bind the EM staining reagent uranyl acetate and appear electron-translucent. Importantly, classical LD-like structures were virtually absent from the double mutant cells. The ER origin of the convoluted membranes present in *pex30Δfld1Δ* and *pex30Δldb16Δ* cells was confirmed by fluorescence microscopy in cells expressing endogenous levels of the ER marker protein Sec63-mNG (Fig. 2d). Shotgun lipidomics analysis[28,29] revealed that, in relation to WT or any of the single mutants, *pex30Δfld1Δ* or *pex30Δldb16Δ* had a dramatic phospholipid surplus consisting

essentially of phosphatidylcholine (PC) and phosphatidylinositol (PI), the major yeast phospholipids (Fig. 2e), which is consistent with the ER expansion phenotype.

The unfolded protein response (UPR) has been shown to control the size of the ER under stress conditions[30,31]. However, basal UPR levels in *pex30Δfld1Δ* or *pex30Δldb16Δ* were comparable to WT cells, suggesting that ER expansion in these cells is UPR-independent (Supplementary Figure 3c). Expanded and convoluted ER membranes have also been observed during ER-phagy, a selective form of autophagy involved in ER turnover[32–34]. However, abrogation of autophagy by deletion of the master regulator kinase Atg1 did not prevent formation of aberrant ER structures in *pex30Δfld1Δ* or *pex30Δldb16Δ* cells (Supplementary Figure 3d). Moreover, in cells expressing the abundant ER marker Sec63-GFP, we did not detect accumulation of free green fluorescent protein (GFP) resulting from autophagic degradation of the ER[32,35] (Supplementary Figure 3e). Thus, the expanded and convoluted ER membranes in *pex30Δfld1Δ* or *pex30Δldb16Δ* appear to originate independently of ER-phagy.

Pex30 was shown to have ER-shaping functions, an activity dependent on its reticulon-homology domain (RHD)[5]. Thus, we wondered whether disruption of major ER-shaping proteins[36], such as the reticulons Rtn1 and Yop1, would also result in defects similar to the ones detected in *pex30Δfld1Δ* and *pex30Δldb16Δ* cells. Remarkably, combined mutations in seipin and the reticulons Rtn1 and Yop1 in *rtn1Δyop1Δfld1Δ* or *rtn1Δyop1Δldb16Δ* cells did not result in Bodipy dispersal in ER membranes or near absence of LDs (Supplementary Figure 3f), suggesting the effects were specific to Pex30. Collectively, these data indicate that combined deletions of seipin and Pex30 block efficient LD budding resulting in trapping of neutral lipids in dramatically enlarged ER.

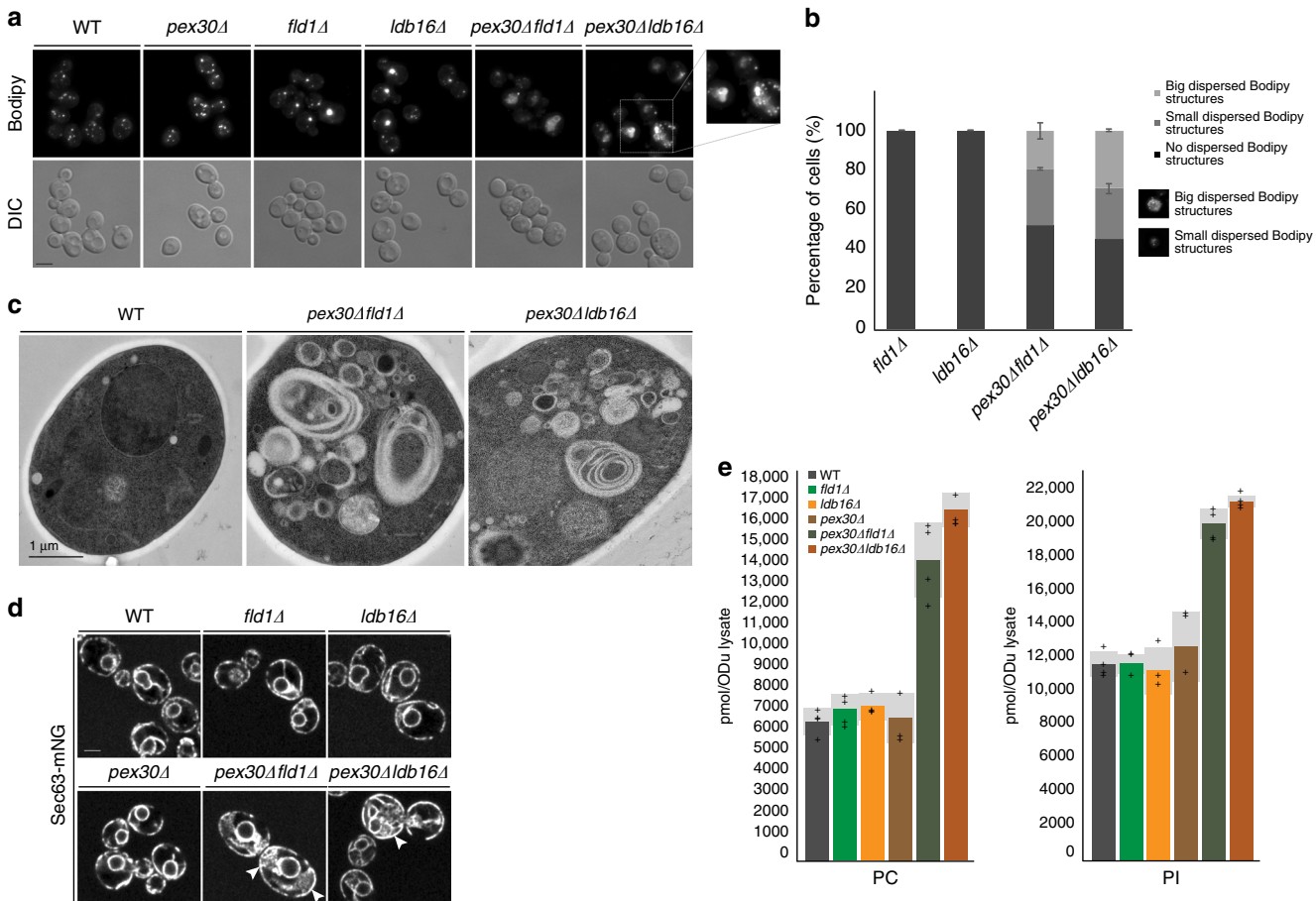

**Fig. 2** Pex30 is required for LD budding in seipin mutants. **a** Analysis of LDs in cells with the indicated genotype after staining with the neutral lipid dye Bodipy493/503. Inset shows aberrant dispersed Bodipy structures detected exclusively in *pex30Δfld1Δ* and *pex30Δldb16Δ* mutants. Bar 2 μm. **b** Quantification of dispersed Bodipy structures in cells with the indicated genotype. Data correspond to the average of two experiments (>100cells/ genotype/experiment were counted). **c** Electron micrographs of WT, *pex30Δfld1Δ*, and *pex30Δldb16Δ* cells. **d** ER morphology in cells with the indicated genotype. ER was visualized by endogenously expressed Sec63-mNG. White arrows show regions with expanded ER. Bar 2 μm. **e** Levels of the major phospholipids phosphotidylcholine (PC, left) and phosphatidylinositol (PI, right) in cells with the indicated genotype, as detected by mass spectrometry. Graphs correspond to the average of two biological and two technical repeats. The four individual measurements are displayed (+). Light gray bars indicate SD

As a first step to dissect the relation between seipin and Pex30 in LD biogenesis, we analyzed their relative distribution. In WT cells growing in glucose-containing media, the vast majority of Fld1-tdTomato foci were proximal to LDs, at ER-LD contacts as previously reported[21,25]. In contrast, only a small fraction of Pex30-mNG foci were proximal to LDs. Strikingly, under conditions favoring LD biogenesis, such as in oleic acid-containing media, both Fld1 and Pex30 were strongly concentrated at ER-LD contacts (Fig. 3). Thus, Fld1 appears to maintain a more stable association with ER-LD contacts, whereas Pex30 associates more transiently, perhaps only during earlier stages of LD biogenesis. Despite the partial colocalization between Pex30 and seipin, we failed to detect a biochemical interaction between these proteins (Supplementary Figure 4). This observation further supports the notion that seipin and Pex30 have partly redundant but independent roles in the budding of LDs from the ER.

**Pex30 and seipin cooperate during peroxisome biogenesis.** The seipin-Pex30 collaboration in LD budding prompted us to test whether these proteins also cooperate in peroxisome biogenesis, in which Pex30 was shown to have a role[5,6,26,37,38]. To follow de novo peroxisome biogenesis, we used a well-established assay

involving regulated expression of Pex3, a protein essential for formation of functional peroxisomes[39]. Upon a GFP-Pex3 pulse, the import of a fluorescent reporter containing a peroxisomal targeting signal (mCherry-PTS1) from the cytoplasm into the matrix of newly formed peroxisomes can be monitored over time by microscopy (Fig. 4a). As previously reported[5], *pex30Δ* mutants displayed a slight delay in peroxisome biogenesis. Mutations in the seipin complex components *fld1Δ* or *ldb16Δ* also resulted in a delay in peroxisome formation. Remarkably, *pex30Δfld1Δ* or *pex30Δldb16Δ* mutants were strongly impaired in de novo peroxisome biogenesis without a single import-competent peroxisome forming during the time course of the experiments (Fig. 4b, c). The defect was not due to different Pex3 levels, as these were similar in all the strains analyzed (Supplementary Figure 5a). Aberrant peroxisome biogenesis in *pex30Δfld1Δ* and *pex30Δldb16Δ* cells was also supported by analysis of mCherry-PTS1 at steady state (Fig. 4d). Although mCherry-labeled peroxisomes were detected in virtually every WT and single mutant cells, only 40% of *pex30Δfld1Δ* and *pex30Δldb16Δ* cells contained peroxisomes (Fig. 4e). In fact, even cells with peroxisomes displayed a detectable cytoplasmic pool of mCherry-PTS1, indicative of compromised import of the matrix protein marker (Fig. 4d).

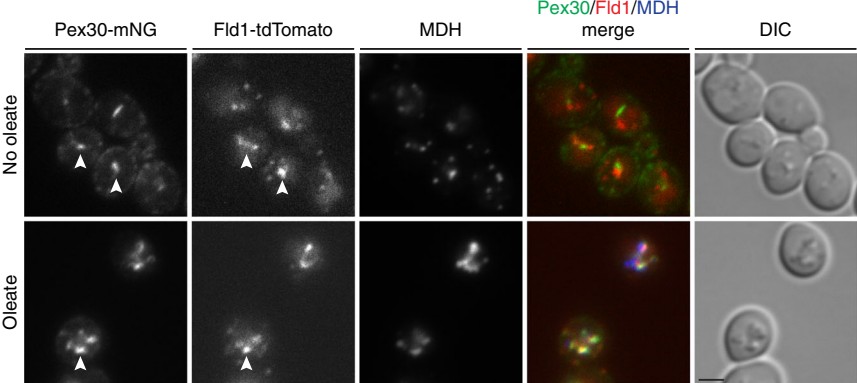

**Fig. 3** Relative distribution of Fld1, Pex30, and LDs. Pex30 and Fld1 were endogenously tagged with monomeric Neon Green (Pex30-mNG) and tdTomato (Fld1-tdTomato), respectively. WT cells were imaged in YPD (no oleate) or in oleate-containing medium. LDs were stained with the neutral lipid dye MDH. White arrowheads show regions where Pex30 is located proximal to Fld1. Bar 2 μm

Consistent with these defects, the prototypical peroxisomal membrane protein, Pex14, a component of the docking complex essential for matrix protein import[40], localized abnormally in *pex30Δfld1Δ* and *pex30Δldb16Δ* cells. By fluorescence microscopy, we detected two to three foci of endogenously expressed Pex14-GFP per cell on each focal plane. In contrast, *pex30Δfld1Δ* and *pex30Δldb16Δ* mutants displayed a diffused pattern with a few interspersed foci overlapping with Sec63-tdTomato, indicating its mislocalization to the ER (Fig. 4f). Thus, although mutations in seipin or Pex30 have mild defects in peroxisome formation, this process is strongly affected in double mutants, suggesting partial redundancy between seipin and Pex30 during peroxisome biogenesis.

Given the striking defects in peroxisomes and LDs, we wondered whether the formation of other ER-derived organelles, such as COPII vesicles would also be affected in cells lacking seipin and Pex30. Remarkably, the ER to Golgi trafficking of carboxypeptidase Y, a well-studied and abundant COPII cargo, was unaffected in *pex30Δfld1Δ* and *pex30Δldb16Δ*, indicating that the effects were specific to LDs and peroxisomes (Supplementary Figure 5b). Altogether, our data indicate that seipin and Pex30 have distinct but partly redundant roles specifically in the formation of LDs and peroxisomes. Moreover, it reveals previously unidentified links in the biogenesis of these structurally unrelated organelles.

**Seipin is enriched at ER sites of PPV budding**. During de novo peroxisome biogenesis, peroxisomal membrane proteins such as Pex3 and Pex14 are initially targeted to the ER, from where they leave in PPVs that eventually mature into peroxisomes[7–9,39]. The budding of PPVs occurs at ER domains enriched in Pex30[5]. Thus, we asked whether seipin components also concentrate at the same ER domains by following de novo peroxisome biogenesis in cells expressing Fld1-tdTomato. Remarkably, at early time points upon induction, GFP-Pex3 almost invariably colocalized with Fld1-tdTomato (Fig. 4g, h). The degree of colocalization decreased over time and was negligible after 90 min (Fig. 4g, h), suggesting complete GFP-Pex3 incorporation into PPVs at this time point. Thus, similar to Pex30[5], seipin is enriched at ER domains involved in PPV budding.

**Loss of Pex30 and seipin leads to toxic TAG accumulation in ER membranes**. So far, we showed that the biogenesis of LDs and peroxisomes from the ER depends on the cooperation of seipin and Pex30. Moreover, in their absence, peroxisomal membrane

proteins such as Pex14 and TAG appear trapped in the ER. Strikingly, essential ER functions in protein trafficking and phospholipid biosynthesis do not appear to be greatly compromised in *pex30Δfld1Δ* and *pex30Δldb16Δ* cells, and are unlikely to be responsible for the growth defect in these mutants. Although these mutants have only a slight increase in TAG (Fig. 5a), it is possible that in the absence of LDs, ER-trapped TAG becomes toxic. To test this hypothesis, we reduced TAG levels in *pex30Δfld1Δ* and *pex30Δldb16Δ* cells by deleting the major TAG-synthesizing enzyme *DGA1*. Remarkably, in *pex30Δfld1Δdga1Δ* and *pex30Δldb16Δdga1Δ* triple mutants, the growth defect was largely suppressed at all temperatures (Fig. 5b). Lipidomics analysis confirmed that *DGA1* deletion indeed resulted in decreased TAG levels in the triple mutants (Fig. 5a). In addition, both the fraction of cells displaying Bodipy dispersal and the size of these structures were decreased in the triple mutants (Fig. 5c, d; Supplementary Figure 6a). In parallel, normal ER morphology was restored in a substantial fraction of cells (Fig. 5e, f), whereas PC and PI dropped to WT levels (Supplementary Figure 6b). Importantly, these changes resulted in a strong alleviation of the peroxisome biogenesis defect (Supplementary Figure 6c). Thus, TAG trapped in ER membranes is toxic and accounts for the slow growth of *pex30Δfld1Δ* and *pex30Δldb16Δ* cells.

**Seipin and Pex30 stabilize ER domains permissive for budding**. Bilayer phospholipid composition and surface tension were recently identified as key factors for the budding of artificial LDs from model membranes in vitro[41] and also appear to regulate LD budding in vivo[42]. Although major global lipidome changes were not detected in cells lacking seipin[4,20,25] or Pex30 (Supplementary Figure 7a), both proteins are capable of affecting membrane properties locally at sites of organelle biogenesis[5,22,43,44]. Thus, an appealing mechanism for seipin and Pex30 to function is through organization of ER domains with lipid composition permissive for organelle budding and that are distinct from bulk ER membranes. In this case, in *pex30Δfld1Δ* and *pex30Δldb16Δ* cells the budding of LDs and PPVs would be dependent on the bulk phospholipid composition of the ER. Indeed, *pex30Δfld1Δ* and *pex30Δldb16Δ* cells have high levels of diacylglycerol (Fig. 6a), a lipid that disfavors positive membrane curvature and strongly inhibits LD budding[41,42]. A second prediction of the model is that conditions generating a more budding-permissive lipid composition should restore organelle budding in *pex30Δfld1Δ* and *pex30Δldb16Δ* mutants. Remarkably, the formation of both LDs (Fig. 6b and Supplementary Figure 7b) and peroxisomes (Fig. 6c, d and Supplementary Figure 7c) were restored in the absence of

seipin and Pex30 upon deletion of Pct1, a rate-limiting enzyme in the Kennedy pathway of phospholipid biosynthesis. In *pex30Δfld1Δpct1Δ* and *pex30Δldb16Δpct1Δ* mutants, the presence of LDs resulted in a concomitant decrease of cells with dispersed Bodipy structures again, suggesting that these correspond to ER-trapped TAG (Supplementary Figure 7d). It should be noted that

LDs in these cells were supersized, similar to those detected in seipin mutants[20,21,25], indicating that besides budding, seipin controls other aspects of LD biogenesis. In these triple mutants, besides lower PC levels, which were expected due to deletion of *PCT1*[45,46], there was a reduction in PI (Fig. 6e) and DAG (Fig. 6a) to levels comparable to WT cells. Concomitantly, there was a

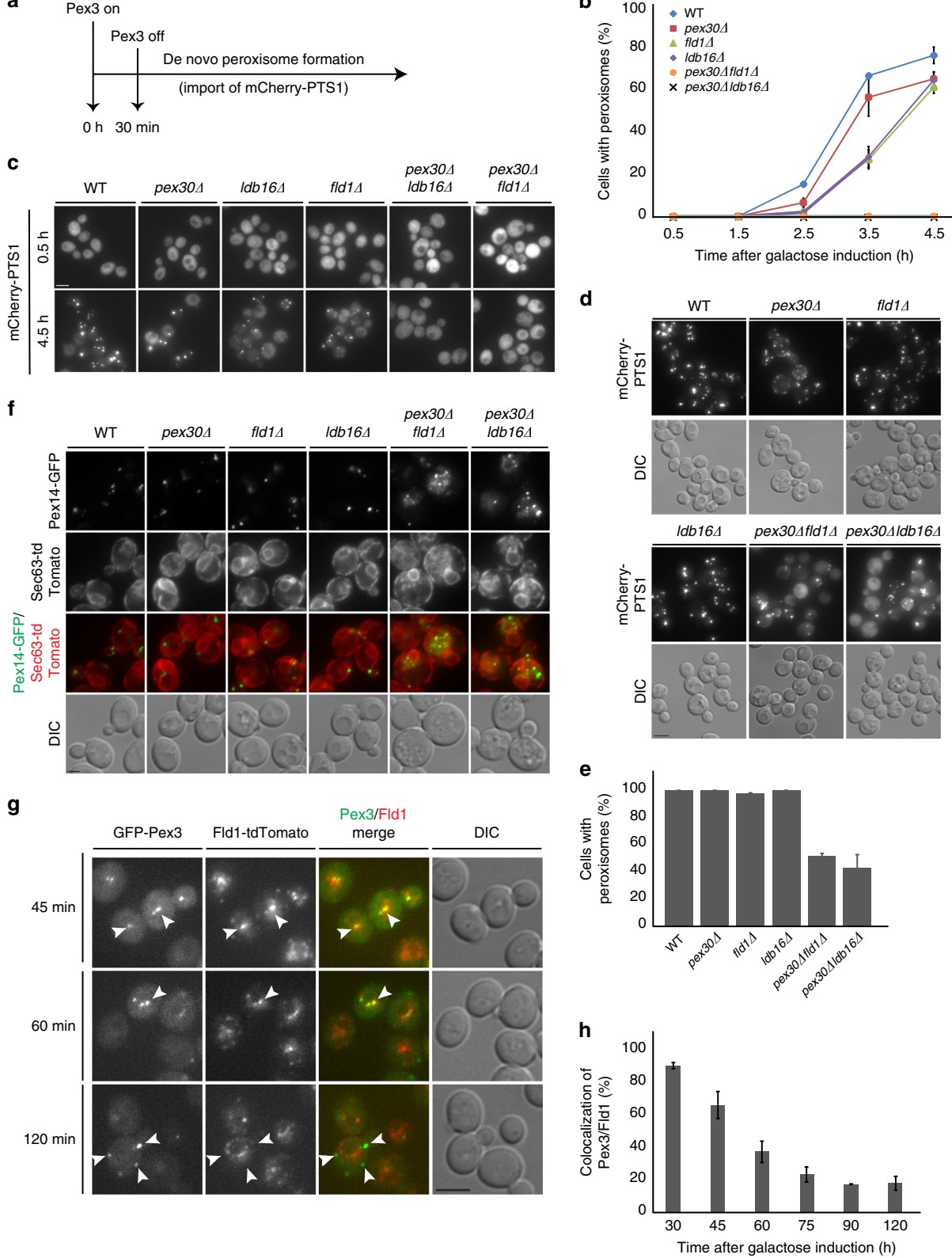

strong increase in TAG levels (Fig. 6f). Thus, in the absence of seipin and Pex30, budding of LDs and PPVs is entirely dependent on ER phospholipid composition.

## Discussion

LDs and peroxisomes are central organelles in cellular metabolism and, in a variety of cell types, are often in close proximity[47–49]. However, the mechanisms by which peroxisomes and LDs form at the ER remain elusive. Here we present evidence that PPVs, which originate peroxisomes, and LDs bud from similar ER domains through a process requiring the cooperation between seipin and Pex30. Simultaneous deletion of these components largely inhibits budding of LDs and PPVs, indicating their partly redundant roles in organelle biogenesis. Remarkably, remodeling of ER phospholipid composition restores organelle budding in the absence of seipin and Pex30, highlighting the role of ER membrane composition in organelle biogenesis.

Curiously, neither seipin nor Pex30 appear to modify lipids directly. In fact, individual deletions of seipin or *PEX30* do not result in major global changes in lipid composition[20,22,50]. Thus, it was puzzling that simultaneous ablation of seipin and Pex30 resulted in dramatic lipidome remodeling, with a strong increase in PC and PI, and to a lower extent, DAG levels. A reduction in TAG synthesis by deletion of *DGA1* largely corrected phospholipid levels in these cells indicating that the defects resulted from inhibition of LD budding and consequent TAG dispersal through ER membranes. Perhaps when trapped in the ER, TAG is more susceptible to lipolysis, resulting in DAG, which accumulates in the ER, as well as in free fatty acids likely shunted to phospholipid synthesis. According to recent in vitro and in vivo studies, DAG is a potent inhibitor of LD budding[41,42]. Given the link between LD and PPV biogenesis uncovered here, we speculate that high amounts of DAG in the ER may also inhibit PPV budding. Consistent with this scenario, deletion of *DGA1* which lowers DAG levels, restores peroxisome formation in the absence of seipin and Pex30.

The abnormal lipid profile of *seipinΔpex30Δ* mutants was also corrected by deletion of *PCT1*, a rate-limiting enzyme of the Kennedy pathway, in which DAG and choline are used to synthesize PC[51]. Thus, a decrease in PC levels was expected in cells lacking Pct1. In contrast, the reduction in DAG is paradoxical as a block in the Kennedy pathway would prevent its consumption. However, concomitant with the reduction of DAG, *seipinΔpex30Δpct1Δ* cells display a strong increase in TAG levels. Thus, by lowering DAG levels and relieving the inhibitory effect of this lipid in organelle budding[41,42], *PCT1* deletion facilitates TAG storage in LDs in *seipinΔpex30Δ* cells. Similarly, the reduction in DAG probably also accounts for the restoration of peroxisome formation in *seipinΔpex30Δpct1Δ* cells. We previously showed that Pct1 relocalizes strongly to LDs in seipin mutants[22], a phenotype also observed in seipin-deficient

*Drosophila* cultured cells[4]. Thus, regulation of Pct1 accumulation and/or activity may be one of the seipin functions at sites of organelle budding.

Pex30 has a RHD with membrane-shaping properties[5] that is required for normal peroxisome biogenesis[6,26,37,38]. Moreover, in its C-terminal cytosolic domain, Pex30 contains a dysferlin-like domain that may bind DAG[38,52,53]. Whether both membrane-shaping and DAG-binding activities are necessary to promote organelle budding in seipin mutants is unclear and should be addressed in the future. However, an appealing possibility is that Pex30 may regulate DAG distribution locally, at sites of organelle biogenesis, an activity that may become essential in the absence of seipin or unregulated Pct1 activity.

Pex30 was also shown to interact with Pex28, Pex29, Pex31 and Pex32, as well as Rtn1 and Yop1[6,37]. The significance of these interactions is not yet clear but it is interesting to note that all these proteins have membrane-shaping properties. However, Pex30-interacting factors do not appear to display genetic interactions with seipin (Supplementary Figure 2 and our unpublished results). Thus, Pex30-mediated organelle budding appears independent of its canonical binding partners. Moreover, ER organelle budding still occurred in cells with combined deletions in seipin and the major ER-shaping proteins Rtn1 and Yop1, suggesting that the defects in *seipinΔpex30Δ* cells are specific and not due to global defects in ER morphology.

How seipin and Pex30 promote organelle biogenesis is not fully understood, but our data supports a model in which these components organize ER domains with a lipid composition distinct from the bulk ER that is permissive for organelle budding. Peroxisomes and LDs are structurally distinct so it will be important to understand how seemingly similar ER sites give rise to monolayered LDs and bilayered PPVs. Despite their differences, both organelles are intimately linked to cellular metabolism and are made in response to similar stimuli[3]. Thus, it is appealing to speculate that the similar requirements during their biogenesis may reflect a common evolutionary origin for these organelles.

## Methods

**Reagents**. GFP antibody (rat monoclonal 3H9; dilution 1:1000) was purchased from Chromotek and PGK1 antibody (mouse monoclonal 459250; dilution 1:10,000) from Invitrogen. Anti-CPY (rabbit polyclonal ab34636; dilution 1:750) was purchased from Abcam. Polyclonal anti-Fld1 (rabbit; dilution 1:3000), anti-Ldb16 (rabbit; dilution 1:5000), anti-Dga1 (rabbit; dilution 1:3000), and anti-Ldo (rabbit; dilution 1:3000) antibodies were previously described[22,54]. Pex30 antibody (rabbit; dilution 1:1000) was a kind gift from William Prinz (National Institute of Diabetes and Digestive and Kidney Diseases, NIH, Bethesda, USA). LD dyes Bodipy493/503 was purchased from Invitrogen and used at a final concentration of 1 μg/ml. Monodansyl pentane (MDH) was purchased from Abgent and used at 0.1 mM.

**Yeast strains and plasmids**. Endogenous protein tagging and promoter replacement were performed using standard PCR-based homologous recombination as described[55,56]. Deletion mutants were made either by PCR-based homologous recombination or by crossing haploid cells of opposite mating types, followed by

**Fig. 4** Pex30 and seipin cooperate during peroxisome biogenesis. **a** Schematic of de novo peroxisome biogenesis induction by pulse of GFP-Pex3 expression, as described[39]. **b** Quantification of de novo peroxisome biogenesis as described in **a** in cells with the indicated genotype. Graphs correspond to the average of two experiments. Error bars indicate the SD. **c** De novo peroxisome biogenesis in cells with the indicated genotype. Representative images of time points 0.5 and 4.5 h are shown. Bar 2 μm. **d** Distribution of functional peroxisomes in cells with the indicated genotype analyzed by steady-state localization of the matrix marker mCherry-PTS1. Bar 2 μm. **e** Quantification of cells of the indicated genotype with functional peroxisomes, as measured by the presence of mCherry puncta corresponding to matrix-imported mCherry-PTS1. The average of two experiments is shown; error bars display standard deviation; >100cells/genotype/experiment were counted. **f** Single focal plane images of cells with the indicated genotype co-expressing endogenously tagged Pex14-GFP and the ER marker protein Sec63-tdTomato. Bar 2 μm. **g** Pex3 and Fld1 transiently colocalize during de novo peroxisome biogenesis. Fld1 was tagged with tdTomato (Fld1-tdTomato) and expressed from the endogenous locus. Representative images of the indicated time point according to **a** are shown. White arrows show bright Pex30 foci that colocalize with Fld1 at 45 min and 60 min, but not 120 min. Bar 2 μm. **h** Quantification of Pex3 and Fld1 colocalization during de novo peroxisome biogenesis, assayed as describe in **a**. The average of two experiments is displayed; error bars represent SD; >100cells/genotype/experiment were counted

sporulation and tetrad dissection using standard protocols[57]. Strains used are isogenic to either BY4741 (*MATa ura3Δ0 his3Δ1 leu2Δ0 met15Δ0*) or BY4742 (*MATα ura3Δ0 his3Δ1 leu2Δ0 lys2Δ0*), and are listed in Table S3.

**Growth conditions**. Cells were grown at 30 °C in YPD liquid medium (1% yeast extract, 2% peptone, 2% glucose) unless indicated otherwise. For experiments using oleic acid, cells were grown to diauxic shift phase in YPD, pelleted, and resuspended in YP + 0.1% oleic acid. For microscopy experiments, exponentially

growing cells were analyzed. For growth assays, cells were grown to exponential phase in the indicated media (either YPD or SC), diluted to an $OD_{600}$ of 0.1 followed by 10-fold serial dilutions. Diluted cells were spotted on either YPD or synthetic complete (SC) plates, and incubated at the indicated temperatures for 2 days.

***De novo* peroxisome biogenesis**. De novo peroxisome biogenesis was analyzed as previously described[39]. In brief, cells were grown in a starter culture in YPD,

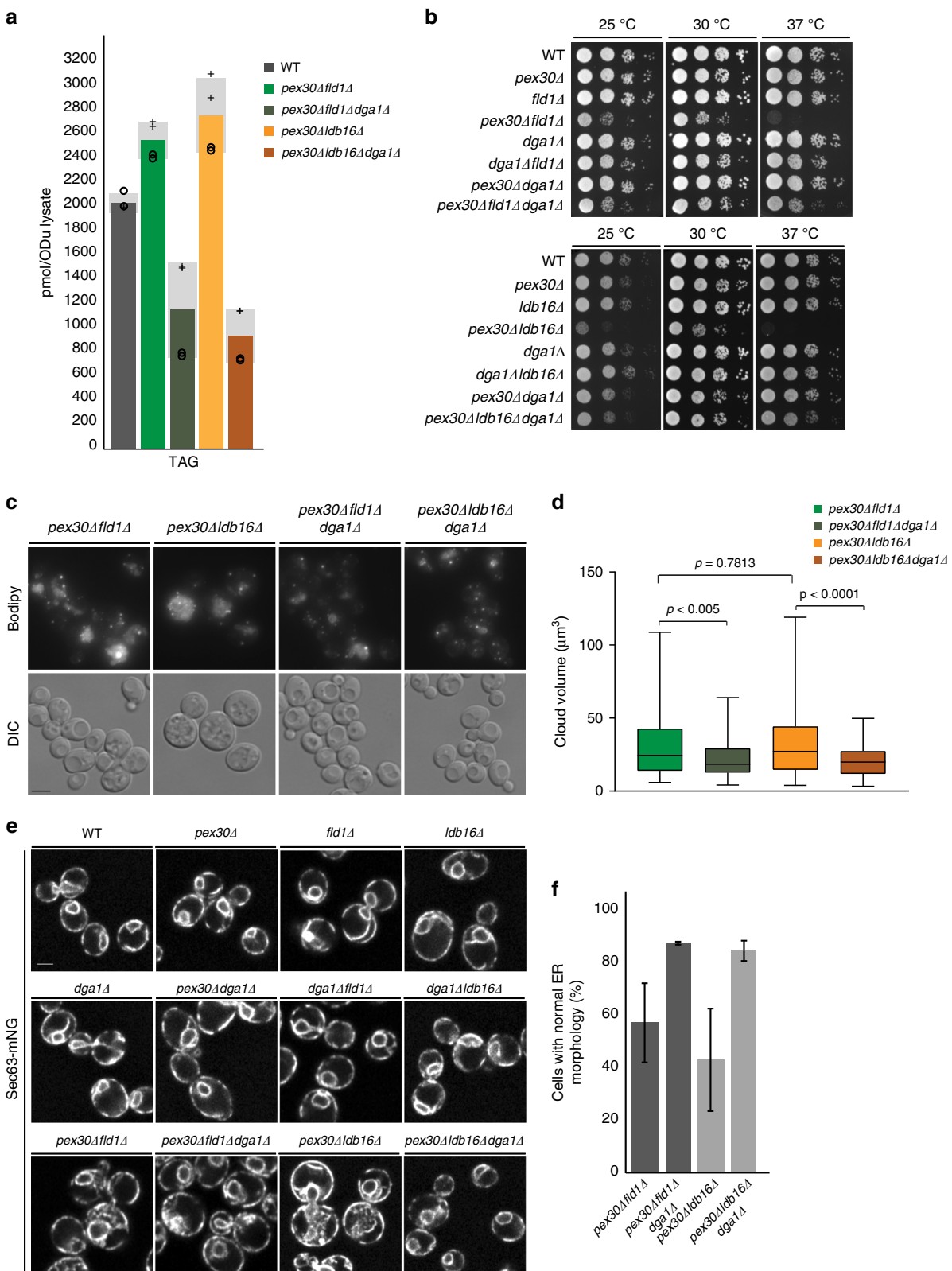

diluted in YP + 4% raffinose, and grown to exponential phase. At this stage, cells were washed and resuspended in YP + 2% galactose. After a 30 min galactose pulse to induce expression of GFP-Pex3, cells were washed, resuspended in YP + 4% glucose, and analyzed at the indicated time points.

**Immunoprecipitation.** Immunoprecipitation of endogenously tagged Fld1-TAP and Ldb16-TAP was performed as previously described[54]. Briefly, 100 $OD_{600}$ units of yeast cultures in log phase were washed and resuspended in 1.4 ml lysis buffer (LB; 50 mM Tris, pH 7.4, 150 mM NaCl, 2 mM $MgCl_2$, and cOmplete protease inhibitor [Roche]). Cells were lysed with glass beads and lysates were cleared by low-speed centrifugation at 4 °C. Membranes were pelleted at 135,000 × g for 25 min at 4 °C in an Optima Max Tabletop Ultracentrifuge in a TLA 100.3 rotor (Beckman Coulter). The crude membrane fraction was resuspended in 600 μl LB. 700 μl of LB supplemented with digitonin to obtain 1% final concentration and membranes were solubilized for 2–3 h on a rotating wheel at 4 °C. Solubilized membranes were cleared for 15 min at 4 °C at full speed in a tabletop centrifuge, and 1.1 ml was used for the immunoprecipitation. TAP -tagged proteins were affinity isolated by overnight incubation with calmodulin sepharose 4B beads (17052901; GE Healthcare). Eluted proteins were analyzed by SDS-polyacrylamide gel electrophoresis (PAGE) and immunoblotting. In all experiments, the input lane corresponds to 5% of the total extract used for immunoprecipitation.

**Pulse-chase and protein analysis.** For pulse-chase analysis, 9 OD units of exponentially growing cells starved for methionine and cysteine were pulse-labeled for 5 min with 250 μCi of [35S] methionine/cysteine followed by addition of cold methionine/cysteine to a final concentration of 2 mM. Collected time points were lysed and CPY was immunoprecipitated with anti-CPY antibody for 2 h at 4 °C. Subsequently, immunocomplexes were retrieved by an additional 1 h incubation at 4 °C with Protein A Sepharose beads. Eluted proteins were separated by SDS-PAGE and visualized by autoradiography. For western blotting, whole-cell extracts of exponentially growing cells were prepared from 2 OD units of cells. Pelleted cells were washed once in phosphate-buffered saline, resuspended in 250 μL of 0.15 M NaOH, and incubated on ice for 10 min. After centrifugation at maximum speed for 2 min at 4 °C, the pellet was resuspended in 1 × sample buffer and heated at 65 °C for 10 min. Proteins were separated by SDS-PAGE in Criterion TGX™ precasted gels (BioRad), transferred to PVDF membrane, and analyzed with the indicated antibodies.

**Fluorescence microscopy.** Most experiments were performed in a Zeiss Cell Observer HS equipped with a CMOS camera (Hamamatsu ORCA-Flash4.0) controlled by 3i Slidebook6.0 software. A Plan-APOCHROMAT 100 × 1.4 objective was used. GFP and BODIPY493/503, mCherry, and MDH signals were detected using GFP, RFP and DAPI filters, respectively, with standard settings. Analysis of ER morphology in Sec63-mNG expressing cells was performed in a Zeiss 880 Airyscan microscope using a 63 × 1.4 Plan-APOCHROMAT objective. All imaging were performed at room temperature.

**Image analysis.** Analysis of dispersed Bodipy structures was performed using 3iSlidebook 6.0 software. Masks were created manually for each dispersed Bodipy structure observed on a single z-position, and volume (in $μ^3$) was measured. Statistical significance and *P*-values were calculated in GraphPad Prism 7 using the Mann–Whitney test.

**Electron microscopy.** Exponentially growing cells were pelleted, loaded into 3 mm, 100 μm, or 200 μm carriers and high-pressure frozen using a Leica EM ICE. Samples were freeze substituted in a Leica AFS2 unit in 1% uranyl acetate +1% water in acetone. Samples were then maintained at − 45 °C for the remainder of the processing. Samples were washed with acetone and then ethanol for 15 min each, then infiltrated with HM20 resin (PolySciences) according to the following schedule: 25% resin in ethanol for 4 h, 50% resin in ethanol overnight, 75% resin in ethanol for 5 h, 100% resin for 6 h, 100% resin overnight, 100% resin for 8 h. Samples were then polymerized with UV light for 48 h, and gradually warmed to 0 °C after the first 24 h of polymerization. 90 nm ultranthin sections were acquired using a Diatome diamond knife with a Leica UC7 ultramicrotome and transferred to formvar-coated 100-mesh copper grids. Sections were post-stained for 10 min with 2% uranyl acetate, washed by passing over a series of warm water droplets,

and then stained with Reynold's lead citrate, washed, and dried. Grids were imaged in a FEI Tecnai 12 TEM operated at 120 kV using a Gatan OneView digital camera.

**LD isolation.** LD purification was carried out as described[22]. Briefly, cells were grown in YPD until stationary phase. Cells (3000 ODs) were centrifuged at 3,000 × g for 5 min (J J26-XP centrifuge and JLA8100 rotor; Beckman Coulter), washed in milliQ water, preincubated in 0.1 M Tris-HCl, pH9.5, and 10 mM dithiothreitol for 10 min at 30 °C, washed, and resuspended in 50 mL spheroplasting buffer (1.2 M sorbitol and 50 mM Tris, pH 7.4). To prepare spheroplasts, Zymolyase 20T (Seikagaku Biobusiness) was added (10 μg/$OD_{600}$ U cells) followed by incubation in a water bath at 30 °C until a 10-fold drop in $OD_{600}$ was observed (45–60 min). Spheroplasts were recovered by centrifuging at 1,000 × g at 4 °C, washed with spheroplasting buffer, and resuspended in breaking buffer (BB: 10 mM MES, Tris, pH 6.9, 12 % [wt/wt] Ficoll 400, 0.2 mM EDTA, 1 mM phenylmethane sulfonyl fluoride and protease inhibitor cocktail). Cells were homogenized using a Dounce homogenizer on ice, and the homogenate centrifuged (5000 × g for 5 min) using rotor JS13.1. The resulting supernatant was centrifuged for 45 min at 175,000 × g (Optima L-100K centrifuge, Beckman Coulter; SW-32 rotor) and the LD-containing floating layer resuspended in BB, overlaid with 19 mL 10 mM MES-Tris, pH 6.9, 8% [wt/wt] Ficoll 400, and 0.2 mM EDTA, and centrifuged as before. The floating layer was collected and resuspended in 10 mM MES-Tris, pH 6.9, 8% [wt/wt] Ficoll 400, 0.6 M sorbitol, and 0.2 mM EDTA, overlaid with 19 mL 10 mM MES-Tris, pH 6.9, 0.25 M sorbitol, and 0.2 mM EDTA, and centrifuged once more at 175,000 × g for 30 min. The highly purified top LD fraction was snap frozen, stored at −80 °C, and used subsequently for proteomics.

**Protein mass spectrometry.** Label-free proteome quantification was carried out as described[22]. Briefly, trichloroacetic acid precipitated proteins were resuspended in 6 M urea and 200 mM ammonium bicarbonate before reduction (10 mM dithiothreitol) and alkylation (20 mM iodoacetamide). Samples were diluted to 2 M urea and digested with trypsin (1:10 wt/wt) overnight at 37 °C. Tryptic peptide mixtures were desalted using a C18 UltraMicroSpin column using three washes with 0.1% formic acid in water, followed by an elution step with 0.1% formic acid in a 1:1 mix of water and acetonitrile[58]. Samples were analyzed in a LTQ-Orbitrap Velos Pro mass spectrometer (Thermo Fisher Scientific) coupled to nano-LC (Proxeon) equipped with a reversed-phase chromatography 12 cm column with an inner diameter of 75 μm, packed with 5 μm C18 particles (Nikkyo Technos). Chromatographic gradients were set from 93% buffer A, 7% buffer B to 65% buffer A, 35% buffer B in 60 min with a flow rate of 300 nl/min (buffer A: 0.1% formic acid in water; buffer B: 0.1% formic acid in acetonitrile). The instrument was operated in DDA mode and full mass spectrometry scans with 1 micro scans at resolution of 60,000 were used over a mass range of m/z 250–2,000 with detection in the Orbitrap. After each survey scan, the top 20 most intense ions with multiple charged ions above a threshold ion count of 5000 were selected for fragmentation at normalized collision energy of 35%. Fragment ion spectra produced via collision-induced dissociation were acquired in the linear ion trap. All data were acquired with Xcalibur software v2.2. Acquired data were analyzed using the Proteome Discoverer software suite (v1.3.0.339; Thermo Fisher Scientific), and the Mascot search engine (v2.3; Matrix Science) was used for peptide identification. Data were searched against database containing all yeast proteins according to the Saccharomyces Genome Database plus the most common contaminants[59]. A precursor ion mass tolerance of 7 ppm at the MS1 level was used, and up to three miscleavages for trypsin were allowed. The fragment ion mass tolerance was set to 0.5 D. Oxidation of methionine and protein acetylation at the N-terminus were defined as variable modifications. Carbamidomethylation on cysteines was set as a fix modification. The identified peptides were filtered using a false discovery rate < 5%. Protein areas were normalized intra- and intersamples by median of the log area. A linear modeling approach implemented in *lmFit* function and the empirical Bayes statistics implemented in *eBayes* and *topTable* functions of the Bioconductor *limma* package[60,61] were used to perform a differential protein abundance analysis. The normalized protein areas of different yeast mutants were compared with *wt* samples. Protein *p*-values were calculated with *limma* and were adjusted with Benjamini–Hochberg method[62]. A value of 0.05 was used as a cutoff.

**Mass spectrometric lipid analysis.** For lipidome analysis cells were grown in YPD until $OD_{600}$ = 2. Lipid extraction and mass spectrometry analysis were done as described previously[28,63]. Briefly, yeast cells were resuspended in 150 nM

**Fig. 5** Absence of Pex30 and seipin results in toxic TAG accumulation in ER membranes. **a** TAG levels in cells with the indicated genotype, as determined by mass spectrometry. Graph corresponds to the average of two biological (+ and ○) and two technical repeats. The four individual measurements are displayed. Light gray bars indicate SD. **b** Tenfold serial dilutions of cells with the indicated genotype were spotted on YPD plates and incubated at 25 °C, 30 °C, or 37 °C for 2 days. **c** Cells with indicated genotype were stained with the neutral lipid dye Bodipy493/503. Bar 2 μm. **d** The volume of dispersed Bodipy structures in cells with the indicated genotype were quantified. Unpaired Mann–Whitney test was used to determine *p*-values. **e** ER morphology in cells with the indicated genotype. ER was visualized by endogenously expressed Sec63-mNG. Bar 2 μm. **f** Quantification of cells with abnormal ER morphology, as determined by microscopy. The average of two experiments is shown; error bars represent SD; >100cells/genotype/experiment were counted

$NH_4HCO_3$ (pH 8) and disrupted by zirconia beads (0.5 mm; Biospec Products). Cell lysates were diluted to 0.2 OD units per 200 µL, and mixed with 20 µL of a cocktail of 20 internal lipid standards. Samples were extracted with 990 µL chloroform/methanol (17:1, V/V) for 120 min. The lower organic 17:1 phase lipid extract was collected. The remaining aqueous sample material was re-extracted with 990 µL chloroform/methanol (2:1, V/V) for 120 min. The lower organic 2:1 phase lipid extract was collected. The lipid extracts were vacuum evaporated. Finally, the lipid extracts were dissolved in 100 µL chloroform/methanol (1:2, V/V). Lipid extracts were analyzed in negative and positive ion mode on a QSTAR Pulsar-i instrument (MDS Analytical Technologies) and a LTQ Orbitrap mass spectrometer (Thermo Fisher Scientific) both equipped with the robotic nanoflow ion source TriVersa NanoMate (Advion Biosciences) as described[64–66]. Detected

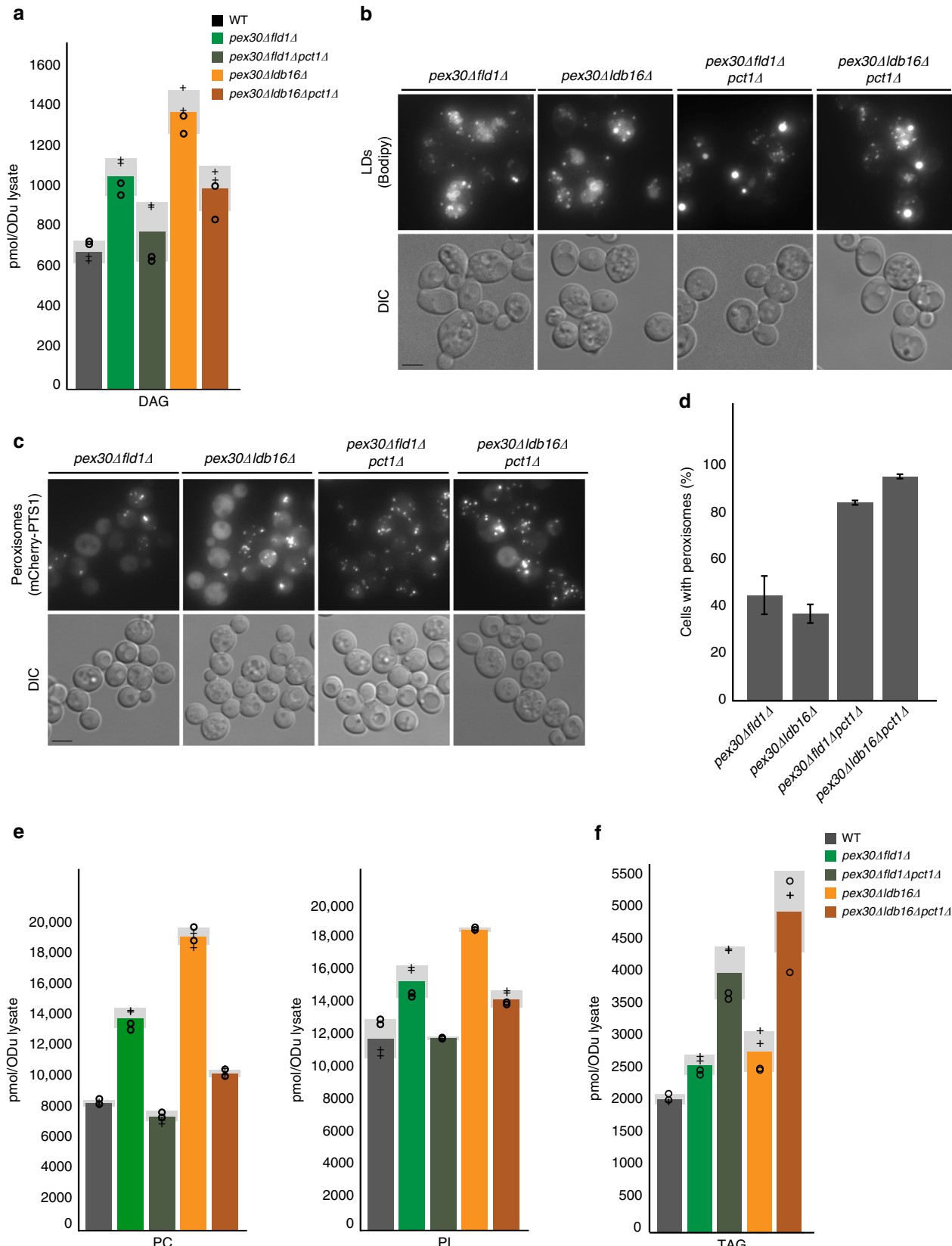

**Fig. 6** Phospholipid composition of bulk ER determines organelle budding in the absence of Pex30 and seipin. **a** DAG levels in cells with the indicated genotype, as determined by mass spectrometry. Graph corresponds to the average of two biological (+ and ○) and two technical repeats. The four individual measurements are displayed. Light gray bars indicate SD. **b** Distribution of Bodipy-stained neutral lipids in cells with the indicated genotypes. Bar 2 μm. **c** Distribution of functional peroxisomes in cells with the indicated genotype analyzed by steady-state localization of mCherry-PTS1. Bar 2 μm. **d** Quantification of functional peroxisomes in cells with the indicated genotype, determined as in **c**. The average of two experiments is shown; error bars represent SD; >100cells/genotype/experiment were counted. **e** Levels of the major phospholipids phosphotidylcholine (PC, left) and phosphatidylinositol (PI, right) in cells with the indicated genotype, as detected by mass spectrometry. Graphs correspond to the average of two biological and two technical repeats. The four individual measurements are displayed (+). Light gray bars indicate SD. **f** TAG levels in cells with the indicated genotype, as detected by mass spectrometry. Graphs correspond to the average of two biological and two technical repeats. The four individual measurements are displayed (+). Light gray bars indicate SD

lipid species were identified and quantified using Lipid Profiler (MDS Analytical Technologies) and proprietary Analysis of Lipid EXperiments (ALEX) software.

**Statistical analysis**. Data for all microscopy experiments were generated from at least two independent experiments. For each experiment, 150 cells were scored per condition and/or genotype from multiple microscopy fields. Distributions are presented as mean ± SD. Where indicated, the volume of dispersed bodipy structures was measured using the 3iSlidebook 6.0 software and statistical significance tested using the Mann–Whitney test in GraphPad Prism 7. Two-tailed P-values are indicated in figures where relevant. Proteomics data result from analysis of six (for WT and ldb16Δ) and two (for fld1Δ) independent LD isolation experiments. Lipidomics data result from two independent biological replicates for all genotypes, each analyzed in two independent technical replicates. In all experiments, cells were randomly selected for analysis and no samples were excluded.

**Data availability**. The data that support the findings of this study are available from the corresponding author upon request.

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

## Acknowledgements

Mass spectrometric protein measurements and data analysis were performed in the Centre for Genomic Regulation (CRG)/Universitat Pompeu Fabra Proteomics Unit, part of the "Plataforma de Recursos Biomoleculares y Bioinformàticos (ProteoRed-Instituto de Salud Carlos III, PT13/0001)". EM samples were prepared at the Dunn School EM Facility with the help of E. Johnson. We thank Sebastian Schuck for reagents, W. Prinz for sharing results before publication, and F. Campelo and the Carvalho Lab for discussions and critical reading of the manuscript. C.S.E. is supported by the VILLUM Foundation (VKR023439) and the VILLUM Center for Bioanalytical Sciences (VKR023179). P.C. is supported by an ERC starting grant, a Wellcome Trust Investigator Award and the EMBO Young Investigator Program.

## Author contributions

:S.W. and F.I. performed most of the experiments and data analysis. M.H. performed lipid extractions, lipid mass spectrometry. and lipidomics data processing with the supervision of C.S.E. A.G. prepared LD-enriched fractions for quantitative proteomics. P. C. conceived and guided the project, analyzed the data, and wrote the manuscript with input from all the authors.

## Additional information

**Competing interests:** The authors declare no competing interests.

