## [Peer Review File · Nature Communications]

Reviewers' comments:

Reviewer #1 (Remarks to the Author):

This study by Wang and colleagues explores a potential collaboration between seipin, a protein complex that stabilizes and regulates the formation of lipid droplets, and Pex30, a protein that regulates the formation of peroxisomes, in the ER. Compared to the classical secretory pathway and the function of COPII and COPI in ER-to-Golgi trafficking, very little is known about the budding requirements of lipid droplets and peroxisomes. Previous work has established a role for seipin in lipid droplet formation and, separately, a role for Pex30 in peroxisome biogenesis. Following up on a proteomic characterization of ER-subdomains that give rise to lipid droplets, Wang and colleagues discover that Pex30 is enriched in these lipid-droplet-forming ER subdomains in seipin deletion mutants. They go on to show that seipin subunits Fld1 and Ldb16 have negative genetic interactions with Pex30 and that each complex localizes not only to sites of lipid droplet formation, but also to sites of peroxisome biogenesis. Using a lipidomics approach, they demonstrate that mutants lacking Pex30 and either one of the seipin subunits have increased phosphatidylcholine and phosphatidylinositol levels. This manifests in a proliferation of the ER into highly convoluted membranes, which the authors confirm in morphological experiments using fluorescence and electron microscopy. Remarkably, this dramatic reorganization of the ER does not influence classical secretion, nor does it upregulate the unfolded protein response of the ER.

The primary advance in this manuscript is the authors' hypothesis that Pex30 and seipin coordinately stabilize membrane domains of lipid composition distinct from the bulk ER and important for the budding of lipid droplets and preperoxisomal vesicles from the ER. In addition to increased PC and PI levels, mutants lacking Pex30 and either one of the seipin subunits have increased diacylglycerol and triacylglycerol levels. The authors nicely demonstrate that deletion of diacylglycerol acyltransferase, Dga1, reduces triacylglycerol levels in Pex30/seipin double mutants and complements the growth defects observed in these strains. Similarly, removal of phosphocholine cytidyltransferase (Pct1), the rate-determining enzyme of the Kennedy pathway for phosphatidylcholine synthesis, lowered levels of phosphatidylcholine but also phosphatidylinositol and diacylglycerol in Pex30/seipin subunit double mutants, complementing their growth defects. Correcting for the accumulation of TAG, or DAG, PC, and PI was sufficient for restoring lipid droplet and peroxisome biogenesis, respectively.

For the most part, this hypothesis seems reasonable; however, I have some concerns over the lack of any specific molecular mechanism(s) that would lend it support. It is clear from the experiments presented that seipin and Pex30 localize to sites of lipid droplet and peroxisome biogenesis, but they themselves are at least one-step removed from both the production of lipids suitable for the biogenesis of these organelles from the ER, and for the actual budding process of lipid droplets and preperoxisomal vesicles themselves. What then, are these proteins doing? Do seipin and Pex30 physically recruit lipid enzymes and budding factors?

Critically, evidence for coordination between Pex30 and seipin is also lacking. Negative genetic interactions aside, how do the authors propose coordination? Are there physical

interactions for these proteins? Or do they recruit the same types of enzymes? A prediction of their hypothesis is that Pex30 and seipin regulate the location and/or activity of proteins such as Pct1 and Dga1. A lack of mechanistic insight for Pex30 and seipin coordination is a major weakness of the manuscript which should be addressed in a resubmission. This may include physical interaction data, or a demonstration of how Pex30 and seipin coordinately regulate effector proteins. A recent report from Joel Goodman's lab shows that Pet10 is critical for lipid droplet assembly and that Dga1 activity is affected in a Pet10 deletion mutant. Similar findings for Pex30 would help the authors' claims that it functions directly in lipid droplet assembly. Also, despite claiming that peroxisomal membrane proteins accumulate in the ER in Pex30/seipin subunit double mutants, there is no evidence presented to support this claim.

In addition, the authors should address these additional concerns:

1. Proposed physiological links between lipid droplets and peroxisomes are not new, or novel. The authors should include references addressing this point, in particular Bins et al. 2006 from Joel Goodman's lab and they should expand discussion to include how the functional physical interactions of these organelles influence their biogenesis and function.
2. In Figure 1 it is unclear whether the proteomics experiments reported in Figure 1a were performed and reported previously in Grippa et al., 2015, or whether they represent new experimental results unique to this manuscript. From the all-to-brief description of the experiment in the Experimental Procedures section I gather that these experiments were performed by Grippa et al. and re-reported here. This should be explicitly stated in the text. In either case, upon reading the materials and methods section of Grippa et al., 2015, it is difficult to ascertain whether biological replicates were performed, or how the steady-state enrichment of Pex30 to sites of lipid droplet formation in stationary cells compares to other constituents of the tubular and cortical ER. The authors should clarify if Pex30 is uniquely enriched, or if it is part of a larger group of enriched proteins. Is the p-value calculated from a single experiment? Previously identified interacting proteins of Pex30 reported by David et al., 2013, and Mast et al., 2016, would be likely candidates, particularly the reticulons. While several abundant ER proteins and Usa1, a non-abundant member of ERAD, are included as negative controls, there is no positive control to demonstrate enrichment of lipid droplet associated proteins. This weakens the claim that Pex30 is uniquely enriched at sites of lipid droplet formation.
3. Pex30 transcript and protein levels are known to vary in response to environmental conditions. However, for most of the experiments reported in this manuscript, the authors do not stick to a consistent growth phase. Some experiments are performed with cells from early log-phase (microscopy) and some from late-log phase (lipidomics and proteomics), which are known to have different requirements for peroxisomes and lipid droplets. Unless there is a justification for why the authors chose to use these different growth conditions, an effort should be made to address this experimentally. At a minimum, it should be acknowledged in the results section as a caveat of their methods.
4. Molecular weight markers are missing from all the western blots presented. It should also be made explicit in the figures that the authors are looking at an endogenously expressed and tagged Pex30, and Pex3, using a GFP antibody. How do they know that the tag hasn't altered the localization and abundance of these proteins?
5. Supplemental Figure 1 is titled incorrectly. It claims that Pex30 enrichment at ER-contact sites is specific and not due to general loss of peroxisomal functions. However, the authors

do not directly test this. The authors claim the negative genetic interaction with seipin subunits is restricted to Pex30. However, careful analysis of supplemental figure 1 shows a subtle interaction for Pex29 and Pex31 for both seipin subunits, and also between Pex32 and Ldb16. It's also apparent that deleting both components of seipin rescues the phenotype. However, this analysis was not included for Pex30. In this figure, Pex19 also appears to have a negative genetic interaction at 25 deg. C. This weakens the claims made in the text. And the authors should look at the triple mutant of Pex30, Ldb16, and Fld1. Should this triple mutant suppress the double mutant phenotypes, it would require a re-evaluation of the authors hypothesis. As a minor concern, the dilution factor should be included in the figures for all spot assays.

6. The authors claim that the reticulons are not involved and present data using an *rtn1/yop1* double mutant to support that claim. However, it well known that loss of Rtn1 is compensated by upregulation of Rtn2 and therefore the conclusion that there is no role for the reticulons is premature. The authors should delete Rtn2 with Rtn1 and Yop1 to rule out a role for the reticulons.

7. The authors use "amorphous clouds" to describe the diffuse distribution of neutral lipids throughout the ER, and in highly convoluted ER membranes. There must be a more suitable and descriptive term that can be used. This term should be removed and replaced with a more appropriate description both in the text and in the figures.

8. The reported string-like appearance of membranes in *pex30delta* is not clearly observed by EM. There is no wild-type reference included in this manuscript for comparison and instead of string-like membranes, there is a cluster of vesicles of unknown origin.

9. The peroxisome biogenesis assay employed in this study may be "well-established"; however, there are major caveats with its use, and these caveats have been acknowledged in the field and addressed extensively in:

a. Motley AM, Galvin PC, Ekal L, Nuttall JM, Hettema EH. Reevaluation of the role of Pex1 and dynamin-related proteins in peroxisome membrane biogenesis. *J Cell Biol.* 2015 Dec 7;211(5):1041-56.

b. Agrawal G, Fassas SN, Xia ZJ, Subramani S. Distinct requirements for intra-ER sorting and budding of peroxisomal membrane proteins from the ER. *J Cell Biol.* 2016 Feb 1;212(3):335-48.

c. Mast FD, Jamakhandi A, Saleem RA, Dilworth DJ, Rogers RS, Rachubinski RA, Aitchison JD. Peroxins Pex30 and Pex29 Dynamically Associate with Reticulons to Regulate Peroxisome Biogenesis from the Endoplasmic Reticulum. *J Biol Chem.* 2016 Jul 22;291(30):15408-27.

d. Agrawal G, Subramani S. De novo peroxisome biogenesis: Evolving concepts and conundrums. *Biochim Biophys Acta.* 2016 May;1863(5):892-901.

At a minimum, these caveats must be acknowledged in the manuscript. Furthermore, it is unclear if the lack of peroxisome formation in the *Pex30/seipin* subunit double mutants arises from a lack of budding in the ER, or if it results from a lack of import of Pex3 into the ER. These two possibilities are not mutually exclusive. The authors claim that peroxisomal membrane proteins accumulate in the ER, and yet the blot in supplementary figure 3 shows similar kinetics for Pex3 in all strains analyzed. If Pex3 isn't going to peroxisomes, and *pex30delta/ldb16delta* cells lack Pex3, where is it going?

Reviewer #2 (Remarks to the Author):

Lipid droplets are important neutral lipid storage organelles that are intimately connected to metabolic diseases such as obesity and diabetes. Although emerging data indicates that lipid droplets bud from the endoplasmic reticulum (ER), the mechanisms involved in lipid droplet biogenesis remain poorly understood. In the current manuscript, the authors employ yeast model systems and discover a fundamental connection between lipid droplet and peroxisome biogenesis. Their data indicate that these two organelles originate from the same ER subdomain and share some biogenesis factors (Fld1/Ldb16 and Pex30). The double deletion experiments reveal a dramatic defect in lipid droplet biogenesis, resulting in triacylglycerol accumulation in the ER and a compensatory increase in PC/PI in ER "clouds". Similarly, the double deletes also display a defect in peroxisome biogenesis.

The discovery of Pex30 as an additional biogenesis factor that cooperates with Fld1/Ldb16 is an exciting new advance with broad implications regarding organelle biogenesis and potential coordinate organelle regulation. Although the mechanism by which Pex30 cooperates with Fld1/Ldb16 in organelle biogenesis is not clear and will require further research, I remain enthusiastic about the publication of this manuscript. Overall, the experiments presented are very rigorous and well controlled. For example, I appreciate the care taken to examine other Pex and reticulon deletion strains, demonstrating that this is a specific relationship of Pex30 with Fld1/Ldb16 and the phenotype is not due to a general defect of ER morphology or peroxisome biogenesis. Similarly, the studies of ER stress and ER protein secretion (CPY) also suggest that these effects are not due to complete ER dysfunction. Actually, the finding that there was no UPR induction was surprising given the expanded ER and the altered ER structure. In brief, this is a strong paper that contributes an important advance to the field and I only have a few minor comments.

Minor comments:

1. There appears to be a difference in the ability of *dga1Δ* to rescue the phenotypes of *pex30Δfld1Δ* and the *pex30Δldb16Δ*. For example, it doesn't appear that *dga1Δ* has much effect on the clouds in *pex30Δfld1Δ*, but it has a pretty strong effect on clouds in *pex30Δldb16Δ*. Is there a reason for this difference? Are the changes in % cells with clouds presented in Fig 4d significant?
2. In figure 3g the red image at 45min does not appear to be the same image used to make the merge. Mistake? In the merge there are red puncta that are absent in the Fld1-tdTomato single channel image.
3. Do Fld1 and Pex30 colocalize?
4. Are these "clouds" related to the "whorls" described by Peter Walter and others? Are the TAG-filled ER membranes undergoing ER-phagy?
5. It would be helpful to include EM micrographs of WT yeast for comparison (Fig 2c).

Reviewer #3 (Remarks to the Author):

One of the big questions that still remains in organelle biology is how peroxisomes and LDs both emerge from the ER. While this has been the ground for many debates there is still no clear molecular understanding. The manuscript by Wang et al. suggests that the yeast Seipin and Pex30 proteins contribute in concert to Peroxisome and LD biogenesis from the ER. They beautifully show that the absence of both factors dramatically reduces the formation of both organelles and that this is dependent on membrane lipid composition. The manuscript explores a timely question in cell biology and the findings are of great interest of the cell biology community. The manuscript is well written, the figures are very esthetic and data are supported by elegant experiments. This is the best manuscript that I have reviewed lately. Hence, I warmly support its publication in Nature Communication.

I would like to suggest minor changes that will strengthen the conclusions and will enable the reader to get all the required information to appreciate the findings:

1. Figure 1b – I suggest adding a third ER marker, at least in the mutant strains, to strengthen the point that Pex30 accumulates in ER-LD contacts.
2. Figure 3g – I suggest adding a third ER marker, at least at an early time point, to strengthen the point that peroxisomes and LD invaginate from the same ER sub-domains.
3. I suggest that all experimental procedures used in the manuscript will be briefly explained in the manuscript, even if they were discussed elsewhere (for example LD isolation and MS LD analysis).

I also suggest moving the description of the de novo peroxisome biogenesis to a separate section (and not as part of the growth condition).

4. I suggest adding a loading control in Sup Figure 3a.
5. I suggest referring to Sup Figure 4a in the text.
6. In the discussion I would be interested to read why the authors find such strong lethality for the Δ seipin/ Δ pex30 combination while loss of neither peroxisomes nor LDs has such an effect on cells. Is it loss of both organelles in parallel or a potential additional role for this protein combination?
7. Some tiny typos are: Line 114 should be “wondered”, Line 117 should be either “mutants” or “a Δ pex30mutant”
8. In experiments (such as in Figure 3e) where standard deviation is shown for two experiments, each of 100 cells, it should be made clear by the author what deviation is statistically measured – of the 200 cells or of 2 repeats?

Reviewer #4 (Remarks to the Author):

In this manuscript, Wang et al. shed new light on the biogenesis of lipid droplets and peroxisomes. They find that Seipin and Pex30 work together and contribute to the budding of these two organelles. Remarkably, they show that the growth defect of the seipin pex30 double mutant can be rescued by deletion of Dga1, nicely demonstrating that failure to sort TAGs from the ER to LDs is toxic for cells. Overall, the experiments performed support the conclusions well and I strongly recommend the publication of this manuscript. I only have a

few minor points.

-In Figure 5A, it not clear why DAG levels decrease in the pex30fld1pct1 triple mutant compared to the pex30fld1 mutant. Intuitively, the expectation is that DAG levels should increase since removing Pct1 actually removes a "consumer" of DAG. The fact that they see a decrease is counterintuitive and the authors might want to comment on it.

-In Figures 4A and 5A, the DAG and TAG levels have been normalized to OD units. This is not ideal because it assumes that refractive properties of the cells and lipid extraction efficiency is equal in all strains. The authors may want to , at least, show that the genetic perturbations have not altered the refractive properties of the cell (e.g. by changing their size or because of the presence of large aberrant structures), by showing that OD units correspond linearly to cell number or cell mass of the mutants analyzed in the two figures.

-Lines 233-235 state that " in pex30 fld1 dga1 and pex30 ldb16 dga1 triple mutants the growth defect was 234 largely suppressed at all temperatures (Fig. 4b)". In Figure 4B, all data corresponding to the fld1 mutant seems however to be missing.

-In line 134, the contrast in EM is provided by uranium and lead. It is a bit imprecise to state that would be more precise to state that the electron translucence results from the presence of non-polar material unable to bind uranyl acetate. In the present form, it reads as if lipids are less electron dense than proteins. This is only true because proteins bind heavy metals.

-In line 113, "we wonder... should read "we wondered"

Reviewer 1:

We thank the reviewer for carefully reading our manuscript and providing constructive comments.

- “It is clear from the experiments presented that seipin and Pex30 localize to sites of lipid droplet and peroxisome biogenesis, but they themselves are at least one-step removed from both the production of lipids suitable for the biogenesis of these organelles from the ER, and for the actual budding process of lipid droplets and preperoxisomal vesicles themselves. What then, are these proteins doing? Do seipin and Pex30 physically recruit lipid enzymes and budding factors? Critically, evidence for coordination between Pex30 and seipin is also lacking. Negative genetic interactions aside, how do the authors propose coordination? Are there physical interactions for these proteins? Or do they recruit the same types of enzymes? A prediction of their hypothesis is that Pex30 and seipin regulate the location and/or activity of proteins such as Pct1 and Dga1”

In the context of this and other studies, we have performed extensive proteomic analysis of seipin complex components as well as Pex30. While we could detect previously identified (David *et al.* MCP 2013; Mast *et al.* JBC 2016; Wang *et al.* JCS 2014) as well as novel (Teixeira *et al.* JCB 2018; Eisenberg-Bord *et al.* JCB 2018) binding partners for these proteins we could not gather evidence that Pex30 interacts with seipin complex. Although a variety of different conditions were used (detergents and buffers) we cannot exclude that these proteins interact very transiently or weakly. These experiments are now described in the text and a representative immunoprecipitation experiment included as supplemental figure.

Our proteomic-based studies of these proteins did not reveal any robust interactions with lipid modifying enzymes. However, we and others previously showed that seipin mutations have a tremendous impact on the composition of the LD proteome. Specifically, we showed that mutations in seipin resulted in aberrant distribution of LD-specific proteins (such as Dga1 and Pet10), which were reduced or heterogeneously distributed to LDs. We also showed that proteins containing amphipathic- helices normally not present at LDs (such as Pct1 or Kes1) were highly enriched in LD-proximal regions. These results are now discussed and referenced in the revised version of the manuscript.

We also analysed the localization of Dga1 and Pet10 in *seipin* Δ *pex30* Δ double mutants, as suggested by the reviewer. The mislocalization of these proteins is already seen in seipin single mutants and is not further modified by additional deletion of Pex30.

Thus, we feel these data does not add much to the paper and decided not to include it in the manuscript. However it is included as a figure for the reviewer.

- “Also, despite claiming that peroxisomal membrane proteins accumulate in the ER in Pex30/seipin subunit double mutants, there is no evidence presented to support this claim.”

We have now re-analyzed the localization of the prototypical peroxisomal membrane protein Pex14-GFP in cells co-expressing the ER marker protein Sec63-tdTomato. The data in Figure 3F shows that in the absence of Pex30 and seipin, Pex14-GFP colocalizes with Sec63-tdTomato, supporting our conclusion that this peroxisomal membrane protein is mislocalized to abnormally expanded ER observed in these mutants.

Other points:

- 1- “Proposed physiological links between lipid droplets and peroxisomes are not new, or novel. The authors should include references addressing this point, in particular Bins et al. 2006 from Joel Goodman’s lab and they should expand discussion to include how the functional physical interactions of these organelles influence their biogenesis and function.”

We now reference previous links between LDs and peroxisomes, including the work of the Goodman lab as well as others.

- 2- “In Figure 1 it is unclear whether the proteomics experiments reported in Figure 1a were performed and reported previously in Grippa et al., 2015, or whether they represent new experimental results unique to this manuscript. From the all-to-brief description of the experiment in the Experimental Procedures section I gather that these experiments were performed by Grippa et al. and re-reported here. This should be explicitly stated in the text. In either case, upon reading the materials and methods section of Grippa et al., 2015, it is difficult to ascertain whether biological replicates were performed, or how the steady-state enrichment of Pex30 to sites of lipid droplet formation in stationary cells compares to other constituents of the tubular and cortical ER. The authors should clarify if Pex30 is uniquely enriched, or if it is part of a larger group of enriched proteins. Is the p-value calculated from a single experiment? Previously identified interacting proteins of Pex30 reported by

David et al., 2013, and Mast et al., 2016, would be likely candidates, particularly the reticulons. While several abundant ER proteins and Usa1, a non-abundant member of ERAD, are included as negative controls, there is no positive control to demonstrate enrichment of lipid droplet associated proteins. This weakens the claim that Pex30 is uniquely enriched at sites of lipid droplet formation.”

The experiments reported in Figure 1A were not previously published in Grippa *et al.* In that study we concentrated on the defects in LD proteome changes in seipin mutants. Here, we concentrate on ER-membrane proteins enriched in LD-associated fractions. We now include data on reticulons, as suggested by the reviewer. Despite being some of the most abundant ER membrane proteins, they are not strongly enriched in these fractions providing further support for the specificity of Pex30. The results and p-values are based on the analysis of 3-5 independent biological replicates.

3- “Pex30 transcript and protein levels are known to vary in response to environmental conditions. However, for most of the experiments reported in this manuscript, the authors do not stick to a consistent growth phase. Some experiments are performed with cells from early log-phase (microscopy) and some from late-log phase (lipidomics and proteomics), which are known to have different requirements for peroxisomes and lipid droplets. Unless there is a justification for why the authors chose to use these different growth conditions, an effort should be made to address this experimentally. At a minimum, it should be acknowledged in the results section as a caveat of their methods.”

The localization of Pex30 has been analyzed during all growth phases and the enrichment in seipin mutants described by us is always observed. It is exclusively dependent on the presence of seipin components and completely independent of the growth phase of the cells. This information is now included in the text.

4- “Molecular weight markers are missing from all the western blots presented. It should also be made explicit in the figures that the authors are looking at an endogenously expressed and tagged Pex30, and Pex3, using a GFP antibody. How do they know that the tag hasn’t altered the localization and abundance of these proteins?”

Molecular weight markers have now been added to all the western blots. Growth assays performed in seipin mutants expressing endogenously tagged Pex30-mNG showed that these strains did not display any defect growing at rates comparable to seipin mutants (or WT cells). Given the growth defect displayed by Pex30 seipin

mutants, these results strongly argue that Pex30-mNG is a functional protein. We now explicitly mention in the text that Pex30-mNG is functional and endogenously expressed.

5- “Supplemental Figure 1 is titled incorrectly. It claims that Pex30 enrichment at ER-contact sites is specific and not due to general loss of peroxisomal functions. However, the authors do not directly test this. The authors claim the negative genetic interaction with seipin subunits is restricted to Pex30. However, careful analysis of supplemental figure 1 shows a subtle interaction for Pex29 and Pex31 for both seipin subunits, and also between Pex32 and Ldb16. It’s also apparent that deleting both components of seipin rescues the phenotype. However, this analysis was not included for Pex30. In this figure, Pex19 also appears to have a negative genetic interaction at 25 deg. C. This weakens the claims made in the text. And the authors should look at the triple mutant of Pex30, Ldb16, and Fld1. Should this triple mutant suppress the double mutant phenotypes, it would require a re-evaluation of the authors hypothesis. As a minor concern, the dilution factor should be included in the figures for all spot assays”

We have corrected the title. The genetic interactions of seipin and different Pex proteins are now presented in Supplemental Figure 2 entitled “Growth defect in cells lacking Pex30 and seipin is specific and not due to general loss of peroxisomal function”. The only consistent and robust genetic interactions we detect with seipin mutants are with Pex30. As pointed out by the reviewer, we detect some minor growth defect of *pex29Δ* and *pex32Δ* with *ldb16Δ*. However, these are very minor and never detected with *fld1Δ*. Thus, we believe they potentially may reflect additional Ldb16 functions outside of the seipin complex.

Similarly, we show that cells lacking seipin together with *PEX3* and *PEX19*, which are essential for peroxisome biogenesis, do not display growth defects as the ones observed with *PEX30*. Similar results were obtained with mutations on other peroxisomal biogenesis essential genes (*PEX5*, 7, 13 and 14). Thus, while we cannot explain the growth defect of the *ldb16Δpex19Δ* mutant (observed exclusively at 25°C), our data strongly support our claim that the growth defect of cells lacking *PEX30* and seipin is not due to general loss of peroxisome function.

Finally, the growth defect of *pex30Δldb16Δ fld1Δ* triple mutant is indistinguishable from both *ldb16Δ pex30Δ* and *fld1Δ pex30Δ* double mutants.

All spotting assays were performed using a 1:10 dilution. This information was originally included in the methods section and is now also in the figure legends.

6- “The authors claim that the reticulons are not involved and present data using an *rtn1/yop1* double mutant to support that claim. However, it well known that loss of Rtn1 is compensated by upregulation of Rtn2 and therefore the conclusion that there is no role for the reticulons is premature. The authors should delete Rtn2 with Rtn1 and Yop1 to rule out a role for the reticulons.”

We disagree with the reviewer. It has been previously demonstrated by the Rapoport group (Voeltz *et al.* Cell 2006) that if expressed alone from its endogenous locus Rtn2 is insufficient to maintain ER morphology. In fact, in the same study it was shown that *rtn1Δ yop1Δ* double mutants display ER morphology defects similar to those of a triple mutant (with *rtn2Δ*). So besides the absence of the bodipy clouds reported in supplemental Figure 2D, we also showed that there was no dramatic enrichment of any of the reticulon-like proteins in LD-associated fractions by mass spectrometry (Figure 1A). Thus, we believe that these data are in line with our conclusion that the effect reported are specific to Pex30 and unlikely to be due to a general ER morphology defect.

7- “The authors use “amorphous clouds” to describe the diffuse distribution of neutral lipids throughout the ER, and in highly convoluted ER membranes. There must be a more suitable and descriptive term that can be used. This term should be removed and replaced with a more appropriate description both in the text and in the figures.”

As mentioned in the text, and acknowledged by the reviewer, the double mutants display a unique bodipy pattern consisting of diffused labeling of the ER with particular accumulation in highly convoluted ER membranes. We believe that the term “amorphous cloud” is a simple and adequate description of the unusual bodipy staining observed in these strains.

8- “The reported string-like appearance of membranes in *pex30delta* is not clearly observed by EM. There is no wild-type reference included in this manuscript for comparison and instead of string-like membranes, there is a cluster of vesicles of unknown origin.”

The text has been edited to more accurately describe the membrane-rich regions sporadically observed in *pex30Δ*. Similar structures have never been detected in wild type cells and a representative image has now been added (Figure 2C).

9- “The peroxisome biogenesis assay employed in this study may be “well-established”; however, there are major caveats with its use, and these caveats have been acknowledged in the field and addressed extensively in:

a. Motley AM, Galvin PC, Ekal L, Nuttall JM, Hettema EH. Reevaluation of the role of Pex1 and dynamin-related proteins in peroxisome membrane biogenesis. *J Cell Biol.* 2015 Dec 7;211(5):1041-56.

b. Agrawal G, Fassas SN, Xia ZJ, Subramani S. Distinct requirements for intra-ER sorting and budding of peroxisomal membrane proteins from the ER. *J Cell Biol.* 2016 Feb 1;212(3):335-48.

c. Mast FD, Jamakhandi A, Saleem RA, Dilworth DJ, Rogers RS, Rachubinski RA, Aitchison JD. Peroxins Pex30 and Pex29 Dynamically Associate with Reticulons to Regulate Peroxisome Biogenesis from the Endoplasmic Reticulum. *J Biol Chem.* 2016 Jul 22;291(30):15408-27.

d. Agrawal G, Subramani S. De novo peroxisome biogenesis: Evolving concepts and conundrums. *Biochim Biophys Acta.* 2016 May;1863(5):892-901. At a minimum, these caveats must be acknowledged in the manuscript. Furthermore, it is unclear if the lack of peroxisome formation in the Pex30/seipin subunit double mutants arises from a lack of budding in the ER, or if it results from a lack of import of Pex3 into the ER. These two possibilities are not mutually exclusive. The authors claim that peroxisomal membrane proteins accumulate in the ER, and yet the blot in supplementary figure 3 shows similar kinetics for Pex3 in all strains analyzed. If Pex3 isn't going to peroxisomes, and pex30delta/ldb16delta cells lack Pex3, where is it going?”

We are aware of ongoing debate on the mechanisms of peroxisome biogenesis and that in the absence of essential peroxisome biogenesis factors, such as Pex3 or Pex1, certain (ghost) vesicles still appear to form. However, I would like to point out that it is now well accepted in the field that, at least under some circumstances, peroxisomal membrane proteins traffic through the ER. Thus, we believe that none of the issues raised in the papers mentioned by the reviewer affect our conclusions. The text includes additional background information on mechanisms of peroxisome biogenesis.

In Supplemental Figure 5a we show that the pulse of Pex3-GFP expression results in equal levels of this protein in all strains. Moreover, we showed that the decay in Pex3-GFP, due to dilution as cells grow and divide, occurs at comparable rates in all strains. This is a control experiment to show that the defects in mCherry-PTS1 import observed in the double mutants are not due to a defect in Pex3 biosynthesis. Considering our data on Pex14 (Figure 4f), we favor the idea that the targeting of peroxisomal proteins to the ER occurs normally in all strains but in the Pex30 seipin double mutants, due to the peroxisomal biogenesis defect, these proteins are mostly retained in abnormally expanded ER membranes.

Reviewer 2:

- 1- “There appears to be a difference in the ability of *dga1Δ* to rescue the phenotypes of *pex30Δfld1Δ* and the *pex30Δldb16Δ*. For example, it doesn’t appear that *dga1Δ* has much effect on the clouds in *pex30Δfld1Δ*, but it has a pretty strong effect on clouds in *pex30Δldb16Δ*. Is there a reason for this difference?”

As the reviewer we also noticed the difference in the extent of the Dga1 rescue in *fld1* and *ldb16* mutant backgrounds. We do not have a plausible explanation for this. While *fld1* and *ldb16* largely phenocopy each other, similar small differences have been in other assays (for example see Grippa et al JCB 2015 Fig 3D). Although we have no evidence for additional functions of Ldb16 we cannot exclude this possibility.

- 2- “In figure 3g the red image at 45min does not appear to be the same image used to make the merge. Mistake? In the merge there are red puncta that are absent in the Fld1-tdTomato single channel image.”

We did check carefully and the images were of the same cell and focal plane. However, to avoid ambiguity, the image was replaced by a different one

- 3- “Do Fld1 and Pex30 colocalize?”

We detected very prominent co-localization of Fld1 and Pex30 in oleate-treated cells, a condition that boosts the production of both peroxisomes and LDs. In the absence of oleate, Fld1 co-localizes only partly with Pex30 foci. This data is now included in Figure 3.

- 4- “Are these “clouds” related to the “whorls” described by Peter Walter and others? Are the TAG-filled ER membranes undergoing ER-phagy?”

To test whether the convoluted ER membranes observed in Pex30 seipin double mutants were undergoing ER-phagy we analysed ER morphology in *atg1Δ* cells which are deficient in autophagy. As shown in Supplemental Figure 3 (panels d and e), Sec63-mNG labelled ER expansions were still detected in *pex30Δ seipinΔ atg1Δ*. Moreover, western blotting did not show ER-phagy mediated accumulation of free GFP in cells expressing Sec63-GFP. Together, these results argue against the

possibility that the convoluted ER membranes observed in *pex30Δ seipinΔ* are related to whorls and result from ER-phagy.

- 5- “It would be helpful to include EM micrographs of WT yeast for comparison (Fig 2c).”

This has been done.

Reviewer 3:

- 1- “Figure 1b – I suggest adding a third ER marker, at least in the mutant strains, to strengthen the point that Pex30 accumulates in ER-LD contacts.”

We thank the reviewer for the suggestion. We replaced the image and the ER marker protein Sec63-mCherry is now included.

- 2- “Figure 3g – I suggest adding a third ER marker, at least at an early time point, to strengthen the point that peroxisomes and LD invaginate from the same ER sub-domains.”

Fld1 localizes permanently to the ER. Moreover, it is well established that under these experimental conditions Pex3-GFP also traffics through the ER. Thus, it is our opinion that a third marker protein is unnecessary. This would also impose a technical complication as we currently do not have any ER marker labelled with CFP, the only available fluorophore.

- 3- “I suggest that all experimental procedures used in the manuscript will be briefly explained in the manuscript, even if they were discussed elsewhere (for example LD isolation and MS LD analysis) I also suggest moving the description of the de novo peroxisome biogenesis to a separate section (and not as part of the growth condition).”

The protocols for LD isolation and MS are now described in further detail.

- 4- “I suggest adding a loading control in Sup Figure 3a”

Supplemental Figure 3A has been replaced by a new experiment including Pgc1 as a loading control.

5- “I suggest referring to Sup Figure 4a in the text”

The figure is now referenced.

6- “In the discussion I would be interested to read why the authors find such strong lethality for the $\Delta seipin/\Delta pex30$ combination while loss of neither peroxisomes nor LDs has such an effect on cells. Is it loss of both organelles in parallel or a potential additional role for this protein combination?”

Our data supports the notion that the growth defect of *pex30 Δ seipin Δ* double mutants results from toxic TAG accumulation in ER membranes. The discussion about this point has now been extended.

7- “Some tiny typos are: Line 114 should be “wondered”, Line 117 should be either “mutants” or “a $\Delta pex30$ mutant”

We thank reviewer for pointing these out. They have been corrected.

8- “In experiments (such as in Figure 3e) where standard deviation is shown for two experiments, each of 100 cells, it should be made clear by the author what deviation is statistically measured – of the 200 cells or of 2 repeats?”

In all cases, the standard deviation measured is between the multiple independent replicates.

Reviewer 4:

1- “In Figure 5A, it is not clear why DAG levels decrease in the *pex30 Δ fld1 Δ pct1 Δ* triple mutant compared to the *pex30 Δ fld1 Δ* mutant. Intuitively, the expectation is that DAG levels should increase since removing Pct1 actually removes a “consumer” of DAG. The fact that they see a decrease is counterintuitive and the authors might want to comment on it”

We agree with the reviewer and we now discuss this issue more explicitly. We speculate that the inefficient LD formation in *pex30 Δ seipin Δ* renders TAG more susceptible to hydrolysis resulting in higher DAG levels in these cells. In contrast, in *pex30 Δ seipin Δ pct1 Δ* cells which are more efficient in LD packaging, TAG is likely to be protected from hydrolysis resulting in lower DAG levels. In agreement with this possibility, we found much higher TAG levels in *pex30 Δ seipin Δ pct1 Δ* cells when compared with the double mutants (Figure 6f).

- 2- "In Figures 4A and 5A, the DAG and TAG levels have been normalized to OD units. This is not ideal because it assumes that refractive properties of the cells and lipid extraction efficiency is equal in all strains. The authors may want to , at least, show that the genetic perturbations have not altered the refractive properties of the cell (e.g. by changing their size or because of the presence of large aberrant structures), by showing that OD units correspond linearly to cell number or cell mass of the mutants analyzed in the two figures."

We understand the reviewer's concerns however we have several indications that the OD measurements do not influence our conclusions. First, OD measurements were used for all the experiments involving protein analysis. In these we did not detect any alterations in the levels of our loading controls suggesting that OD measurements provide a reasonable measure of cell number. Second, the lipid changes are specific to the indicated species in each case and not global again arguing against changes in cell refractive properties. Finally, cells were spiked with exogenous lipid standards for assessment of lipid extraction efficiency and absolute quantitation. These measurements provide results similar to the OD measurements.

- 3- "Lines 233-235 state that " in pex30 fld1 dga1 and pex30 ldb16 dga1 triple mutants the growth defect was 234 largely suppressed at all temperatures (Fig. 4b)". In Figure 4B, all data corresponding to the fld1 mutant seems however to be missing."

The data has now been added.

- 4- "In line 134, the contrast in EM is provided by uranium and lead. It is a bit imprecise to state that would be more precise to state that the electron translucence results from the presence of non-polar material unable to bind uranyl acetate. In the present form, it reads as if lipids are less electron dense than proteins. This is only true because proteins bind heavy metals."

The text has been edited for accuracy.

- 5- "In line 113, "we wonder... should read "we wondered""

This has been fixed.

a**b**
Reviewers' comments:

Reviewer #1 (Remarks to the Author):

In this revised manuscript, Wang and colleagues provide evidence that the budding of lipid droplets and preperoxisomal vesicles occurs from subregions of the ER enriched for seipin components Fld1 and Ldb16, and Pex30. Cells lacking Pex30 in combination with either Fld1 and Ldb16 have altered ER phospholipid metabolism. Adjusting ER phospholipid composition with removal of genes involved in these metabolic pathways restores defects in lipid droplet and peroxisome biogenesis. The conclusion is that seipin and Pex30 cooperatively organize subregions of the ER permissive for the budding of nascent lipid droplets and peroxisomes. The manuscript is much improved from its original submission; however, the authors were unable to provide mechanistic insight into how Pex30 and seipin coordinate their function. This is not without effort on their part; the authors made good faiths effort to identify physical interactions between Pex30 and seipin, and for the ability of seipin and Pex30 to alter the localization of key enzymes such as Pct1 and Dga1. Despite a definitive mechanism the authors present compelling evidence for the main conclusions of the manuscript – “that seipin and Pex30 act in concert to organize membrane domains permissive for organelle budding”. Nonetheless, some of the conclusions deserve tempering. The data for Pex30 and seipin in lipid droplet formation is more compelling than the data for their role in peroxisome formation. Specifically, the evidence for the accumulation of peroxisomal membrane proteins in the ER in double mutants of Pex30 and seipin components is relatively weak (see below). Therefore, the manuscript would benefit from careful consideration of the points below, and appropriate revision of the text.

1. Given that no physical interactions could be detected, and no distinguishable phenotypes could be observed for the localization of Pct1 and Dga1, the conclusions made by the authors are too strongly stated. As I mentioned, toning down the language used to describe the authors favoured hypothesis would be beneficial.
2. The authors have addressed some of my concerns with the inclusion of experimental methods and additional data for their proteomics data, but several concerns remain:
 - a. Although requested in the prior review, a positive control to demonstrate that they are measuring protein content from a lipid droplet enriched fraction is lacking. Grippa et al., showed enrichment for Opi1 and Pct1, among others, in fld1 and ldb16 null mutants. Also, the authors should comment on why they didn't detect Pex30 in their previous measurements in Grippa et al., but now do.
 - b. The authors have provided insufficient data and information for properly evaluating these proteomics data. Given that “3-5” biological replicates have been performed, a measure of error must be included in the table in Figure 1A. A table summarizing which samples were measured 3 times and which samples were measured 5 times should be included in the supplemental information.
 - c. Without having access to the full dataset, it is difficult to appreciate how unique the results for Pex30 are compared to other unreported proteins, in particular Pex28, Pex29, Pex31 and Pex32. As a condition of acceptance, the authors should deposit the full proteomics dataset in PRIDE and provide the appropriate accession numbers in the manuscript, as is consistent with Nature Communications guidelines.

3. Differences in growth between wt and pex30delta are indistinguishable in the assays used by the authors and we don't know what mechanistic function Pex30 has in peroxisome or lipid droplet biogenesis. Therefore, the claim that Pex30-GFP is functional (Page 5, line 108) is overstated and incorrect.
4. Citations on Page 6, line 127 are incorrect - should be Vizeacoumar et al., 2003 and 2004.
5. Based on supplemental figure 2 there are genetic interactions occurring between seipin subunits and pex29, pex31 and pex32. These are alleviated by deleting both seipin subunits. The authors must acknowledge this in the text. These interactions are not as dramatic as those measured between Pex30 and seipin subunits, but they are present.
6. Page 6, lines 131-135. Peroxisomes are not required by yeast in glucose medium. What function are the authors proposing is compromised here?
7. The authors do not provide data on a pex30/fld1/ldb16 triple null mutant. In their response to the previous review they state that the triple null mutant grows at rates comparable to the double mutants. As mentioned in the previous review, this triple null mutant is important, especially in light of the alleviating phenotypes observed for the other Pex30 family members in this background. I strongly recommend including the pex30/fld1/ldb16 triple null mutant in the manuscript. Minimally, the authors should include a description of the growth phenotype for this triple null mutant in the text.
8. Page 7 line 151. These EM data support, but do not confirm the observations made by fluorescence microscopy as intensity of the bodipy-positive structures is not measured by EM.
9. Page 7 line 155. The authors are intent on using "clouds" to describe the diffusion of neutral lipids in ER membranes. This language lacks precision and the manuscript would benefit from a more descriptive term. If this were my manuscript I would want to use language that was unambiguous in order to aid comprehension for both the specialist and more general audience.
10. Page 7 line 164. I recommend "translucent" instead of "transparent".
11. Page 8 lines 176-190. I recommend making a more cautious conclusion from the experiments described here. Instead of claiming no role for UPR and/or ER-phagy I would leave room for noncanonical roles. This is because there is no positive control included for these experiments and the mutants tested are not comprehensive. It may be that a novel form of UPR and/or ER-phagy is involved and if this is the case it is likely non-canonical.
12. Page 8 lines 192-203. The authors do not test the rtn1/rtn2/yop1 triple mutant despite appreciating the fact that the RHD of Pex30 was discovered in an overexpression screen in a rtn1/rtn2/yop1 null background strain (Joshi et al. 2016). In response to the comments of the prior review, the authors argue that Rtn2 plays no role in defining the structure of the general ER. This may well be a valid point that remains for further validation outside the scope of this manuscript; however, that does not exclude a role for Rtn2 in this context lipid droplet and peroxisome biogenesis, which the authors show is limited to a subdomain of the ER. Absence of evidence for a role for Rtn2 in general ER morphology is not evidence of absence of a role for Rtn2 in defining the ER subdomain necessary for lipid droplet and peroxisome biogenesis.
13. Page 10 line 237. How do the authors know that these punctate fluorescent structures formed in this assay in wild-type or single null mutants are functional peroxisomes? Functional and nonfunctional peroxisomes are not distinguishable when grown in rich

medium containing glucose.

14. Page 10 lines 242-245. How do the authors know that the lack of peroxisomes is a biogenesis defect as opposed to an inheritance defect? This is not mutually exclusive as Pex30 has been suggested to also have roles peroxisome inheritance through its interaction with Inp1.

15. Page 10 lines 248-255. The authors only detect 2-3 GFP-Pex14 per cell, which likely indicates that this fluorescently tagged protein has dominant-negative effects and is possibly non-functional. Therefore, caution should be used when interpreting the localization of this reporter in pex30/fld1 and pex30/ldb16 null mutants. Furthermore, these measurements are made from cells at steady state and not induced for biogenesis. Therefore, the mislocalization observed, in addition to the dominant negative effect of the GFP fusion construct, cannot be attributed to defects in de novo biogenesis alone. The conclusions from this experiment should be tempered. The authors cannot conclude that peroxisomal membrane proteins are accumulating in the ER (as stated here, in the abstract, and in the discussion), based on this experiment.

16. Page 13 line 311-312. The toxicity of TAG trapped in ER membranes seems inconsistent with COPII release that is indistinguishable from wt, unless the accumulated TAG is asymmetrically retained in the mother cell during division. While outside the scope of this manuscript, the authors may want to investigate whether such asymmetry does indeed exist and, whether replicative lifespan is reduced in these mutants. Should such an inheritance mechanism be dependent on Pex30, this would be one alternative mechanism for explaining the observations made here.

17. Page 17 lines 410-420. This paragraph is incorrect. Pex30-interacting factors do have genetic interactions with seipin and the full extent of the interaction between Pex30 and seipin, and the reticulons and seipin, remain unexplored by the authors. This paragraph should be rewritten to reflect these points.

18. Figure 4 and Supplementary Figure 7 use the same DIC and fluorescent image for WT expressing mCherry-PTS1. This should be corrected.

19. Supplementary Figure 4 has Ldb16 misspelled as Ldo16 in the second from the top panel.

Reviewer #2 (Remarks to the Author):

The authors have addressed all of my concerns and I support publication. As previously stated, this is an interesting and exciting manuscript that provides new insights into lipid droplet and peroxisome biogenesis. It will be of broad interest and impact.

Reviewer #3 (Remarks to the Author):

The authors have now answered all of my comments and suggestions in a satisfactory manner and I now strongly support publication of this important work in Nature Communications.

Response to Reviewer 1:

We were pleased to see the reviewer found our manuscript much improved. We are also thankful for the careful analysis and constructive comments.

1. “Given that no physical interactions could be detected, and no distinguishable phenotypes could be observed for the localization of Pct1 and Dga1, the conclusions made by the authors are too strongly stated. As I mentioned, toning down the language used to describe the authors favoured hypothesis would be beneficial.”

We thank reviewer for the suggestion. We slightly modified the discussion on the potential interplay between Pex30 and Pct1 and how these may regulate lipid domains involved in organelle budding. We also softened some of the conclusions, as suggested.

2 “The authors have addressed some of my concerns with the inclusion of experimental methods and additional data for their proteomics data, but several concerns remain:

a. Although requested in the prior review, a positive control to demonstrate that they are measuring protein content from a lipid droplet enriched fraction is lacking. Grippa et al., showed enrichment for Opi1 and Pct1, among others, in *fld1* and *ldb16* null mutants. Also, the authors should comment on why they didn't detect Pex30 in their previous measurements in Grippa et al., but now do.

b. The authors have provided insufficient data and information for properly evaluating these proteomics data. Given that “3-5” biological replicates have been performed, a measure of error must be included in the table in Figure 1A. A table summarizing which samples were measured 3 times and which samples were measured 5 times should be included in the supplemental information.

c. Without having access to the full dataset, it is difficult to appreciate how unique the results for Pex30 are compared to other unreported proteins, in particular Pex28, Pex29, Pex31 and Pex32. As a condition of acceptance, the authors should deposit the full proteomics dataset in PRIDE and provide the appropriate accession numbers in the manuscript, as is consistent with Nature Communications guidelines.”

In Grippa et al. we used several methods to show that the fractions analyzed by mass spectrometry were strongly enriched in lipid droplets. As mentioned before, on that manuscript we focus on the defects in LD proteome changes in seipin mutants, in particular on the reduction of LD-specific proteins and increase of amphipathic-helix containing proteins. Here, we concentrate on ER-membrane proteins enriched in those LD-associated fractions. We are now including two additional tables (S1 and S2) listing all the proteins that were significantly changed ($p < 0.001$) in *fld1Δ* and *ldb16Δ* mutants in comparison to LDs isolated from WT cells. These include the hits described in Grippa et al. as well as Pex30, which is the focus of this study. More information is now included on the number of independent biological replicates (6 for WT and *ldb16Δ*; 2 for *fld1Δ*).

3. “Differences in growth between wt and *pex30Δ* are indistinguishable in the assays used by the authors and we don't know what mechanistic function Pex30 has in peroxisome or lipid droplet biogenesis. Therefore, the claim that Pex30-GFP is functional (Page 5, line 108) is overstated and incorrect.”

The reviewer is correct that the growth of *pex30Δ* is indistinguishable from WT. But as we mentioned in our previous response to the reviewer, we tested the functionality of Pex30-mNG in a seipin mutant background. In this context, while *pex30ΔseipinΔ* has a growth defect (as shown in Fig 1d and Sup Fig2a), *PEX30-mNG seipinΔ* grow at the same rate of WT and *seipinΔ* single mutants. This functionality assay is included in the Figure for the reviewer. We believe that based on this genetic complementation experiment it is fair to conclude that Pex30-mNG is functional.

4. “. Citations on Page 6, line 127 are incorrect - should be Vizeacoumar et al., 2003 and 2004.”

This has been fixed.

5. “Based on supplemental figure 2 there are genetic interactions occurring between seipin subunits and *pex29*, *pex31* and *pex32*. These are alleviated by deleting both seipin subunits. The authors must acknowledge this in the text. These interactions are not as dramatic as those measured between Pex30 and seipin subunits, but they are present.”

As we mentioned in our previous letter the only consistent and robust genetic interactions we detect with seipin mutants is Pex30. Very minor defects have been observed for *pex29Δ*, *pex31Δ* and *pex32Δ* in some of the spotting assays. However, these are inconsistent and of a magnitude difficult to measure given the sensitivity of the assay. We have now included a figure for the reviewer with multiple replicates of the spotting assay. Finally, we believe that these data are described in a balanced manner (we say that, besides Pex30, deletion of other family members in seipin mutant background grow mostly at normal rates).

6. “Page 6, lines 131-135. Peroxisomes are not required by yeast in glucose medium. What function are the authors proposing is compromised here?”

The text has been edited for accuracy.

7. “The authors do not provide data on a *pex30/fld1/ldb16* triple null mutant. In their response to the previous review they state that the triple null mutant grows at rates comparable to the double mutants. As mentioned in the previous review, this triple null mutant is important, especially in light of the alleviating phenotypes observed for the other Pex30 family members in this background. I strongly recommend including the *pex30/fld1/ldb16* triple null mutant in the manuscript. Minimally, the authors should include a description of the growth phenotype for this triple null mutant in the text.”

A new growth assay including the *pex30Δfld1Δldb16Δ* triple mutant has now been included (Fig 1d). As mentioned previously, the growth of this strain is indistinguishable from the individual *pex30ΔseipinΔ* double mutants. As can be seen in this and other growth assays included in the figure for the reviewer, the phenotypes of mutants with deletion in one or both seipin complex subunits are indistinguishable. As mentioned in point 5, we cannot exclude that very minor differences exist but in that scenario they are very small and below the sensitivity of the assay.

8. "Page 7 line 151. These EM data support, but do not confirm the observations made by fluorescence microscopy as intensity of the bodipy-positive structures is not measured by EM."

The text has been edited for accuracy.

9. "Page 7 line 155. The authors are intent on using "clouds" to describe the diffusion of neutral lipids in ER membranes. This language lacks precision and the manuscript would benefit from a more descriptive term. If this were my manuscript I would want to use language that was unambiguous in order to aid comprehension for both the specialist and more general audience."

We removed the term "cloud" and followed the reviewer's suggestion to describe this unusual structures.

10. "Page 7 line 164. I recommend "translucent" instead of "transparent"."

The change has been done.

11. "Page 8 lines 176-190. I recommend making a more cautious conclusion from the experiments described here. Instead of claiming no role for UPR and/or ER-phagy I would leave room for noncanonical roles. This is because there is no positive control included for these experiments and the mutants tested are not comprehensive. It may be that a novel form of UPR and/or ER-phagy is involved and if this is the case it is likely non-canonical."

In yeast, there is vast literature showing that Ire1 is the major player in UPR activation and Atg1 has been shown to be necessary for ER-phagy (Mochida et Nature 2015). Thus, based on current knowledge of the yeast UPR and ER-phagy, we believe that our conclusions are not overstated. Obviously, "there are also unknown unknowns", however these are more difficult to discuss.

12. "Page 8 lines 192-203. The authors do not test the *rtn1/rtn2/yop1* triple mutant despite appreciating the fact that the RHD of Pex30 was discovered in an overexpression screen in a *rtn1/rtn2/yop1* null background strain (Joshi et al. 2016). In

response to the comments of the prior review, the authors argue that Rtn2 plays no role in defining the structure of the general ER. This may well be a valid point that remains for further validation outside the scope of this manuscript; however, that does not exclude a role for Rtn2 in this context lipid droplet and peroxisome biogenesis, which the authors show is limited to a subdomain of the ER. Absence of evidence for a role for Rtn2 in general ER morphology is not evidence of absence of a role for Rtn2 in defining the ER subdomain necessary for lipid droplet and peroxisome biogenesis.”

We agree with the reviewer that the putative role of Rtn2 in stabilization of specialized ER domains is outside of the scope of this manuscript.

13. “Page 10 line 237. How do the authors know that these punctate fluorescent structures formed in this assay in wild-type or single null mutants are functional peroxisomes? Functional and nonfunctional peroxisomes are not distinguishable when grown in rich medium containing glucose.”

The text has been edited for clarity.

14. “Page 10 lines 242-245. How do the authors know that the lack of peroxisomes is a biogenesis defect as opposed to an inheritance defect? This is not mutually exclusive as Pex30 has been suggested to also have roles peroxisome inheritance through its interaction with Inp1.”

Based on steady state distribution of mCherry-PTS1 we indeed cannot rule out a potential effect on peroxisome inheritance. However, the data in Figs 4b-c as well as the defect in mCherry-PTS1 import at steady state (Fig 4d) are all indicative of a defective biogenesis. In any case we softened our conclusion on this point.

15. “Page 10 lines 248-255. The authors only detect 2-3 GFP-Pex14 per cell, which likely indicates that this fluorescently tagged protein has dominant-negative effects and is possibly non-functional. Therefore, caution should be used when interpreting the localization of this reporter in pex30/fld1 and pex30/ldb16 null mutants. Furthermore, these measurements are made from cells at steady state and not induced for biogenesis. Therefore, the mislocalization observed, in addition to the dominant negative effect of the GFP fusion construct, cannot be attributed to defects in de novo biogenesis alone. The conclusions from this experiment should be tempered. The authors cannot conclude that peroxisomal membrane proteins are accumulating in the ER (as stated here, in the abstract, and in the discussion), based on this experiment.”

We would like to point out to the reviewer that the Pex14-GFP images in Figure 4F correspond to single focal planes (and not to z maximal intensity projections) explaining the low number of Pex14-GFP foci. This information has now been added to the text and figure legend. Single planes illustrate better the dispersed distribution of

Pex14-GFP and its co-localization to the aberrant ER membranes. We tagged the endogenous chromosomal PEX14 with GFP, a strategy that is the gold standard to monitor protein localization and dynamics. A similar approach was used by others investigating Pex14 (for example Joshi et al. JCB 2016; Knoops et al. JCB 2015) and it is well established that Pex14-GFP co-localizes with other bona fide peroxisomal proteins. In total, we normally detect 3-5 brighter and several dimmer and smaller Pex14-GFP spots in each cell, which has also been reported by others. Moreover, the pattern and distribution of Pex14-GFP and mCherry-PTS1 is very similar when a Z maximal intensity projection is made again suggesting that Pex14-GFP is a good surrogate of Pex14. Thus, we do not see any evidence for a dominant negative effect of the tagged protein and the co-localization with tdTomato-Sec63 in *pex30ΔseipinΔ* strongly support our conclusion that the Pex14-remains in the ER these mutants.

16. "Page 13 line 311-312. The toxicity of TAG trapped in ER membranes seems inconsistent with COPII release that is indistinguishable from wt, unless the accumulated TAG is asymmetrically retained in the mother cell during division. While outside the scope of this manuscript, the authors may want to investigate whether such asymmetry does indeed exist and, whether replicative lifespan is reduced in these mutants. Should such an inheritance mechanism be dependent on Pex30, this would be one alternative mechanism for explaining the observations made here."

We would like to thank reviewer for the suggestion. The differential effect of *pex30ΔseipinΔ* mutation on LD and PPVs when compared to COPII vesicles is certainly something of interest to us and that we are further investigating. We are certainly testing whether aberrant ER structures are compartmentalized from the remaining, more "functional" ER. As suggested by the reviewer it will be interesting to see whether these domains are asymmetrically partitioned during cell division.

17. "Page 17 lines 410-420. This paragraph is incorrect. Pex30-interacting factors do have genetic interactions with seipin and the full extent of the interaction between Pex30 and seipin, and the reticulons and seipin, remain unexplored by the authors. This paragraph should be rewritten to reflect these points."

As discussed in points 5 and 7, we do not have very strong evidence for a growth defects in the strains in question. However, we worded our conclusions in a more careful manner.

18. "Figure 4 and Supplementary Figure 7 use the same DIC and fluorescent image for WT expressing mCherry-PTS1. This should be corrected."

Those images are part of the same experiment (as well as the ones for *pex30Δ*, *fld1Δ* and *ldb16Δ*). However, to avoid confusion we have now replaced the images in Figure 4D.

19. "Supplementary Figure 4 has Ldb16 misspelled as Ldo16 in the second from the top panel."

There is no misspelling in Supplementary Figure 4. The second panel of that figure shows a blot against Ldo proteins (Ldo45 and Ldo16) which are generated by a splicing event and share the C-terminal portion recognized by the same antibody. The functions of these proteins in LD regulation were recently described by us (Teixeira et al. JCB 2018) and the Schuldiner lab (Eisenberg-Bord et al. JCB 2018).

Growth assays in *pex28Δ*, *pex29Δ*, *pex31Δ* and *pex32Δ*

Functionality assay for Pex30-mNG